# A molecular gradient along the longitudinal axis of the human hippocampus informs large-scale behavioral systems

Jacob W. Vogel[1]*, Renaud La Joie [2], Michel J. Grothe [3], Alexandr Diaz-Papkovich[4,5], Andrew Doyle [1], Etienne Vachon-Presseau[6,7], Claude Lepage[1], Reinder Vos de Wael[1], Rhalena A. Thomas[1], Yasser Iturria-Medina[1], Boris Bernhardt[1], Gil D. Rabinovici[2] & Alan C. Evans[1]*

The functional organization of the hippocampus is distributed as a gradient along its longitudinal axis that explains its differential interaction with diverse brain systems. We show that the location of human tissue samples extracted along the longitudinal axis of the adult human hippocampus can be predicted within 2mm using the expression pattern of less than 100 genes. Futhermore, this model generalizes to an external set of tissue samples from prenatal human hippocampi. We examine variation in this specific gene expression pattern across the whole brain, finding a distinct anterioventral-posteriodorsal gradient. We find frontal and anterior temporal regions involved in social and motivational behaviors, and more functionally connected to the anterior hippocampus, to be clearly differentiated from posterior parieto-occipital regions involved in visuospatial cognition and more functionally connected to the posterior hippocampus. These findings place the human hippocampus at the interface of two major brain systems defined by a single molecular gradient.

---

[1] Montreal Neurological Institute, McGill University, Montréal, QC, Canada. [2] Memory and Aging Center, University of California, San Francisco, CA, USA. [3] German Center for Neurodegenerative Diseases (DZNE), Rostock, Germany. [4] McGill University and Genome Quebec Innovation Centre, Montréal, QC, Canada. [5] Quantitative Life Sciences, McGill University, Montreal, QC H3A 0G1, Canada. [6] Faculty of Dentistry, Department of Anesthesia, McGill University, Montréal, QC, Canada. [7] Alan Edwards Centre for Research on Pain (AECRP), McGill University, Montréal, QC, Canada. *email: jacob.vogel@mail.mcgill.ca; alan.evans@mcgill.ca

The hippocampus is a phylogenetically conserved and well-connected structure involved in a diverse multitude of behaviors, providing an excellent base for studying the evolution of cognition. Alongside its highly nuanced and well-documented role in memory, the hippocampus has been implicated in many other behaviors and functions, ranging from social cognition to spatial orientation to regulation of endocrine processes, such as stress response[1,2]. Commonly studied as a uniform structure, the hippocampus can be divided into well-described subfields—the cornu ammoni (CA), dentate gyrus, and subiculum—which represent its principal axis of organization, and which strongly inform cytoarchitectonic variation and both internal and external circuitry[1]. A second orthogonal axis of organization of the hippocampus lies along its longitudinal axis in a gradient spanning its two poles. In the rodent, this axis is often referred to as the ventral–dorsal axis, while a homologous gradient is thought to exist in humans along the anterior–posterior axis[3–5]. To study variations along this axis, the hippocampus is often divided into basic macroscopic partitions; the head–body–tail division is often used in humans, whereas a dorsal–ventral division is used in rodents. The divisions along the longitudinal axis of the hippocampus are characterized by a complex but distinct pattern of afferent and efferent connections, as well as impressive behavioral domain specificity. In rodents, the ventral hippocampus shares connections with the prefrontal cortex, basolateral amygdala, hypothalamus, and other structures mediating neuroendocrine and autonomic signaling and motivated behavior. Meanwhile, the dorsal hippocampus is anatomically connected with retrosplenial cortex, mamillary bodies, anterior thalamic complex, and other networks implicated in movement, navigation, and exploration[2,4]. Studies directly assessing the existence of a homologous longitudinal organizational axis in the human hippocampus have found compelling evidence in support[6–9], and evidence has emerged suggesting this axis defines the multifaceted role of the hippocampus in complex cognitive systems[10] and in vulnerability to neurodegenerative diseases[11,12].

Centrally involved in so many aspects of brain function and dysfunction, a comprehensive study of the hippocampus and its organizational principles may be paramount to understanding the brain at large. With this concept in mind, several studies have explored the molecular properties that vary along the longitudinal axis of the hippocampus. A number of studies have characterized the genomic anatomy of the ventral–dorsal axis of the rodent hippocampus as a whole or across specific subfields[13–17], how gene expression along the axis changes over the course of development[18,19], and how it influences patterns of connectivity[2]. While some consensus over implicated genes has been met, all of these studies have been performed exclusively in rodents, and it is unclear whether similar genes and proteins are involved in regulating and characterizing the anterior–posterior axis of the human hippocampus. This distinction is important, as the human hippocampus bears a different anatomy from that of rodents, participates in ostensibly more complicated cognitive systems, and shows selective vulnerability to diseases unique to humans.

As yet, such explorations have been severely limited due to the complications of measuring regionally detailed gene expression in the human brain. However, the Allen Human Brain Atlas[20] has provided unprecedented access to human brain gene expression data. In the current study we leverage gene expression data to define the genomic anatomy of the longitudinal axis of the human hippocampus. Specifically, we sought to understand whether, as with the rodent hippocampus, notable gene expression variations also exists along the human hippocampus, and which genes are most prominently involved in this molecular organization. We further aimed to understand whether information about gene expression can help explain interactions between the hippocampus and the diverse brain systems it participates in, as well as differential vulnerability to neurodegenerative disease. To accomplish this, we drew from several public and private human datasets to bridge molecular properties with brain structure and function, behavior, and finally, dissociated vulnerability to neurodegenerative disease. We show that a graduated pattern of gene expression along the hippocampal longitudinal axis predicts the location of a brain tissue sample along this axis, and that distinct interactions between the anterior and posterior hippocampus with specific brain systems can be predicted by the genomic similarity shared between those brain systems and the different poles of the hippocampus.

## Results

**Sparse gene sets predict sample location along the hippocampus.** Normalized gene expression information from 58,692 probes were obtained from each of 170 brain samples extracted from the hippocampi of six deceased human donors from the Allen Human Brain Atlas. The longitudinal axis of the hippocampus, from the anterior to the posterior pole, was defined as a curve passing through the center of mass of the hippocampal volume of an average brain template in MNI standard space. The position of each of the 170 hippocampus samples was projected onto this longitudinal axis (Fig. 1a, Supplementary Fig. 1b). A LASSO-PCR algorithm was used to create a model predicting the position of each sample based on its gene expression profile (Supplementary Fig. 1).

Using repeat ten-fold cross-validation, the LASSO-PCR model explained 68–73% of the variance in sample position along the longitudinal axis (average MAE = 2.17 mm) using only gene expression information (Fig. 1b, c).

Hippocampus samples were extracted from six different subfields as labeled by the Allen Brain anatomist: CA 1–4, the dentate gyrus, and the subiculum. By training our model on five subfields and then using this model to predict the position of the sixth left-out subfield (i.e. leave-one-subfield-out), we revealed that the genomic signature underlying the anterior–posterior gradient of the hippocampus is consistent across hippocampal subfields (Fig. 1d), though the variance predicted was poorer for CA2 ($r^2 = 0.47$) and the subiculum ($r^2 = 0.58$) compared to CA1, CA3, CA4, and the dentate gyrus ($r^2$s > 0.73). Leave-one-donor-out prediction additionally suggested consistency of the genomic signature across individuals (Fig. 1e): while two donors accounted for over 60% of the samples, when samples from these two donors were included in the model, prediction of the location of samples for the other four donors was highly accurate ($r^2$s > 0.80).

Weights from the LASSO-PCR model were back-transformed onto the individual probes in order to highlight the variation of individual genes along the hippocampal longitudinal axis. Weights from L1-regularized regression (LASSO) are difficult to reliably interpret[21], making identification of individual candidate genes challenging. To circumvent this issue, we iteratively removed the probes with 50 highest (anterior-associated) and 50 lowest (posterior associated) weights, respectively, refit the model, and measured cross-validation accuracy of the new model, until all 58,692 probes were removed (Fig. 1f). Removing the first set of 100 probes (Set 1) resulted in a sharp drop in cross-validation accuracy that was never recovered, suggesting that this gene set is particularly important to the model. Accuracy dropped once again after removing the next 500 probes (Set 2; rank 101–600), and after the next 1100 probes were removed (Set 3; rank 601–2700), cross-validation accuracy began to drop precipitously, finally bottoming out after another 2100 probes (Set 4; rank 2700–4800) were removed (Fig. 1f, g). In contrast, iteratively removing sets of 100 random probes resulted in a very

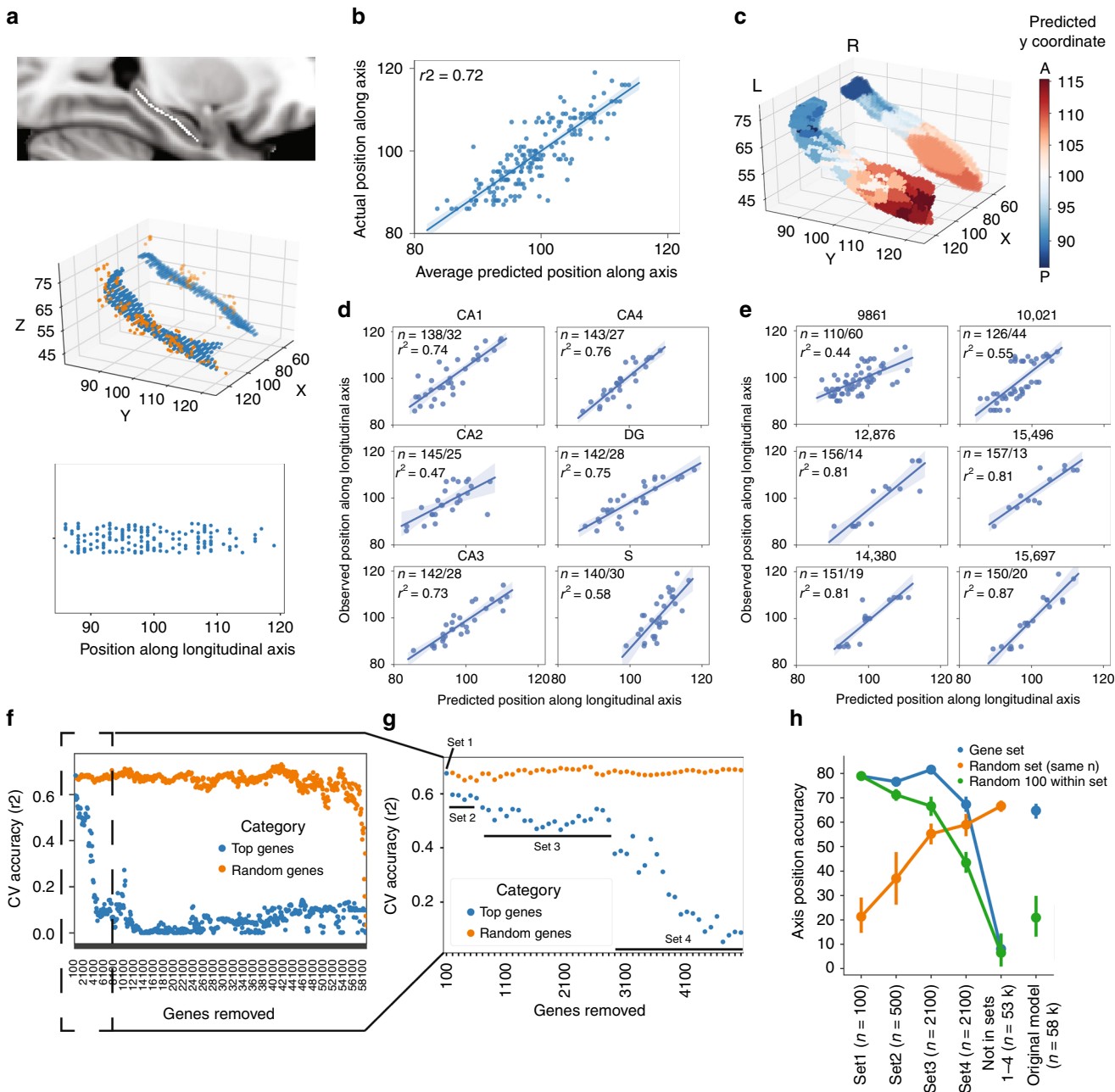

**Fig. 1 Gene expression predicts the location of tissue samples along the long axis of the hippocampus. a** (top) A curved skeleton of voxels was fitted along the center of mass of the hippocampal volume. (middle) Tissue samples (orange) were matched to the closest skeleton voxels (blue). (bottom) A sample's position along the longitudinal axis was represented as the y-axis coordinate of the sample's matched skeleton-voxel. **b** Average predicted sample position (using gene expression) across ten separate 10-fold cross-validated LASSO-PCR models, compared to the actual position. **c** Render of the hippocampal surface where each vertex shows the predicted location of the closest (surface projected) sample to that vertex. The smooth appearance of the right hippocampus is related to the fact that less samples were available for this structure. **d** Predicted vs. observed sample locations for leave-one-subfield-out models. For example, subpanel CA1 shows the predicted vs. observed position of samples extracted from CA1 (test set) when the model was trained without CA1 samples (training set). In each plot, N represents the number of samples in the training and test sets. **e** Predicted vs. observed sample locations for leave-one-donor-out models. **f** The 100 most important probes in the LASSO-PCR model were iteratively removed and, after each removal, 10-fold cross-validation accuracy predicting sample position along the longitudinal axis was recorded (blue dots). As a control, the same process was repeated but removing 100 random probes (orange). **g** The first 50 rounds of 100-probe removal from Panel F. Inflection points were identified after removing 100, 600, and 2700 genes. **h** Accuracy in predicting sample position was recorded for models using different gene sets identified by the inflection points in panel G (blue), samples of 100 random within-set probes (green), and samples of random probes (orange) as input. Each model was run ten times with different bootstrap samples to calculate confidence intervals (represented by error bars).

**Table 1 The top anterior- and posterior-associated genes, respectively, identified by the LASSO-PCR model.**

| Anterior gene | Prev. reported | Posterior gene | Prev. reported |
|---|---|---|---|
| AQP3 | | BDKRB1 | |
| C1QL1 | | BNC2 | |
| C1orf187 | | C1orf133 | |
| C20orf103 | | CASR | |
| CD36 | | COL5A2 | |
| GABRQ | | CTXN3 | |
| GDA | 16,18,19 | DDC | |
| GPR26 | | DGKI | 18 |
| GPR39 | 18 | FAM43B | |
| GPR83 | | FSTL4 | 18 |
| GPR88 | | GAL | |
| KCNG1 | 13 | GEFT | |
| KIAA1772 (GREB1I) | | GREM2 | |
| KLK7 | | GRHL2 | |
| LGALS2 | | HHIP | |
| LMO1 | | HPSE2 | |
| LXN | | KDELR3 | |
| LYPD1 | 18 | NPNT | |
| MYB | | NTN1 | |
| NR2F2 | 13,16,18,19 | ONECUT2 | |
| NRG1 | | OSBPL3 | |
| OPRK1 | | PDLIM5 | |
| PIRT | | PVALB | |
| PYDC1 | | RGMA | |
| RSPH9 | 18 | SERTAD4 | 13,18 |
| RSPO2 | 13 | TNNT2 | |
| SEMA3D | | TPBG | 14 |
| SERPINF1 | | TTR | 16 |
| SSTR1 | 14,18,19 | WNT10A | |
| SYTL1 | | | |
| SYTL2 | | | |
| TMEM215 | | | |
| VGLL3 | | | |

The top 100 probes and their respective beta coefficients can be found in Supplementary Data 1.

gradual and sporadic decrease in accuracy that only bottomed out when nearly all probes were removed (Fig. 1f). Refitting the LASSO-PCR model with only probes from Set 1 (100 probes), Set 2 (500 probes) or Set 3 (2100 probes) resulted in cross-validation accuracy above 80% (MAE: Set 1 = 1.84 mm; Set 2 = 2.39 mm; Set 3 = 1.85 mm), a substantial improvement over the original model and a considerable improvement over models with equal-sized sets of random genes. Genes from Set 4 (2100 probes) alone achieved accuracy similar to a model using all (58,692) probes, and a model using all 53,892 probes not included in Sets 1–4 achieved cross-validation accuracy near 0% (Fig. 1h). These results indicate that 100 specific probes are sufficient to accurately predict the location of a sample along the longitudinal axis of the hippocampus, and that probes outside of a specific set of 4800 provide little to no information about the axis. Fitting the model using gene Sets 2 and 3 alone resulted in cross-validation accuracy similar to Set 1, suggesting the possibility that important genes may also be present within these feature sets. However, the accuracy may also be assisted by the larger number of features included in these two sets. Indeed, random sets of 100 features taken from within Sets 2 and 3 showed reduced cross-validation accuracy compared to Set 1 and full Sets 2 and 3 (Fig. 1h).

**Genes associated with the long axis of the human hippocampus.** A list of the 100 top probes (Set 1) can be found in Supplementary Data 1. This set of probes was associated with a total of 61 genes, which are listed in Table 1. This gene set also

included several genes previously identified to differentiate the dorsal and ventral aspects of the rodent hippocampus (see Table 1). Without exception, all genes found here to be preferentially expressed in the anterior hippocampus were also found to be expressed in the ventral hippocampus of these previous rodent studies, and vice versa for the posterior hippocampus. Gene ontology (GO) enrichment analysis of the top 100 probes from the model revealed a consistent set of terms relating to regulation of anatomical structure morphogenesis and tissue (particularly axonal) growth and development. (Fig. 2a). Among this gene set, a feature explainer based on cross-validated Random Forest Regression suggested NR2F2 and RSPH9 as, on average, the most important local predictors of position along the longitudinal hippocampus axis (Fig. 2e). This result remained consistent when additionally adding all probes from Sets 2 and 3 (Supplementary Fig. 2). In addition to NR2F2 and RSPH9, the feature explainer also implicated local contributions to individual samples from FAM43B, FSTL4, and NTN1 (Fig. 2e). The expression pattern of these five genes differed, as each pattern likely added unique information to the model (Fig. 2f). Feature explainers run on Sets 2 and 3 alone revealed more contributing features with less individual importance, compared to Set 1 and pools including Set 1 (Supplementary Fig. 2).

To further explore the specific patterns of gene expression across the longitudinal axis, all genes across Sets 1, 2, 3, and 4 were entered into a clustering algorithm. Fourteen distinct anterior–posterior patterns emerged, including highly linear patterns (clusters 3 and 6), some highly non-linear patterns (Clusters 7 and 9), and a number of step-wise or step-gradient patterns (Supplementary Fig. 3). Unsurprisingly, Set 1 was composed of a higher proportion of linear expression patterns (Fig. 2c, d), and Set 1 probes within less-linear clusters tended to have more linear expression patterns than average (Fig. 2d), and than other sets (Supplementary Fig. 3). Sets 2–4 therefore may contain a mix of genes expressed in a gradient along the longitudinal axis, along with genes that are specifically hyper-expressed in the anterior or posterior hippocampus. This suggests individual sample predictions are likely aided by different genes depending on their location along the longitudinal axis. Cluster membership of each probe can be found in Supplementary Data 2.

We also performed GO enrichment analysis on all genes represented in Set 2, and then clustered genes sharing similar enrichment terms (Supplementary Data 3). One cluster emerged sharing similar terms to those enriched in Set 1, relating to regulation of axon guidance, as well as cell motility, migration and development. This cluster also included genes previously described in studies exploring the rodent longitudinal axis, including SLIT2, IGFBP5, JUN, and CADM1. Other GO enrichment sets included amine metabolic processes, GABA receptor activity, hormonal signaling, neuropeptide receptor activity, ion transport, and serotonin receptor activity. These latter gene clusters may be more likely to regulate behaviors differentially associated with the anterior or posterior hippocampus. We repeated this analysis for Set 3 (Supplementary Data 4). Once again, a cluster of genes emerged associated with cell motility and migration, which again included genes previously described from the rodent literature (e.g. NTNG2, SEMA3E, NOV, SEMA4G, CADM1, CYP26B1).

As a way of validating the candidate genes identified in our model, we repeated our analyses using Partial Least Squares regression (PLSR), another algorithm appropriate given the high dimensionality of our data. Using all probes, we obtained similar overall cross-validation results (Supplementary Fig. 4). Of the top 100 probes identified by the PLSR model (Supplementary Data 1), 50 were included in Set 1, another 42 in Set 2, and the last 8 were

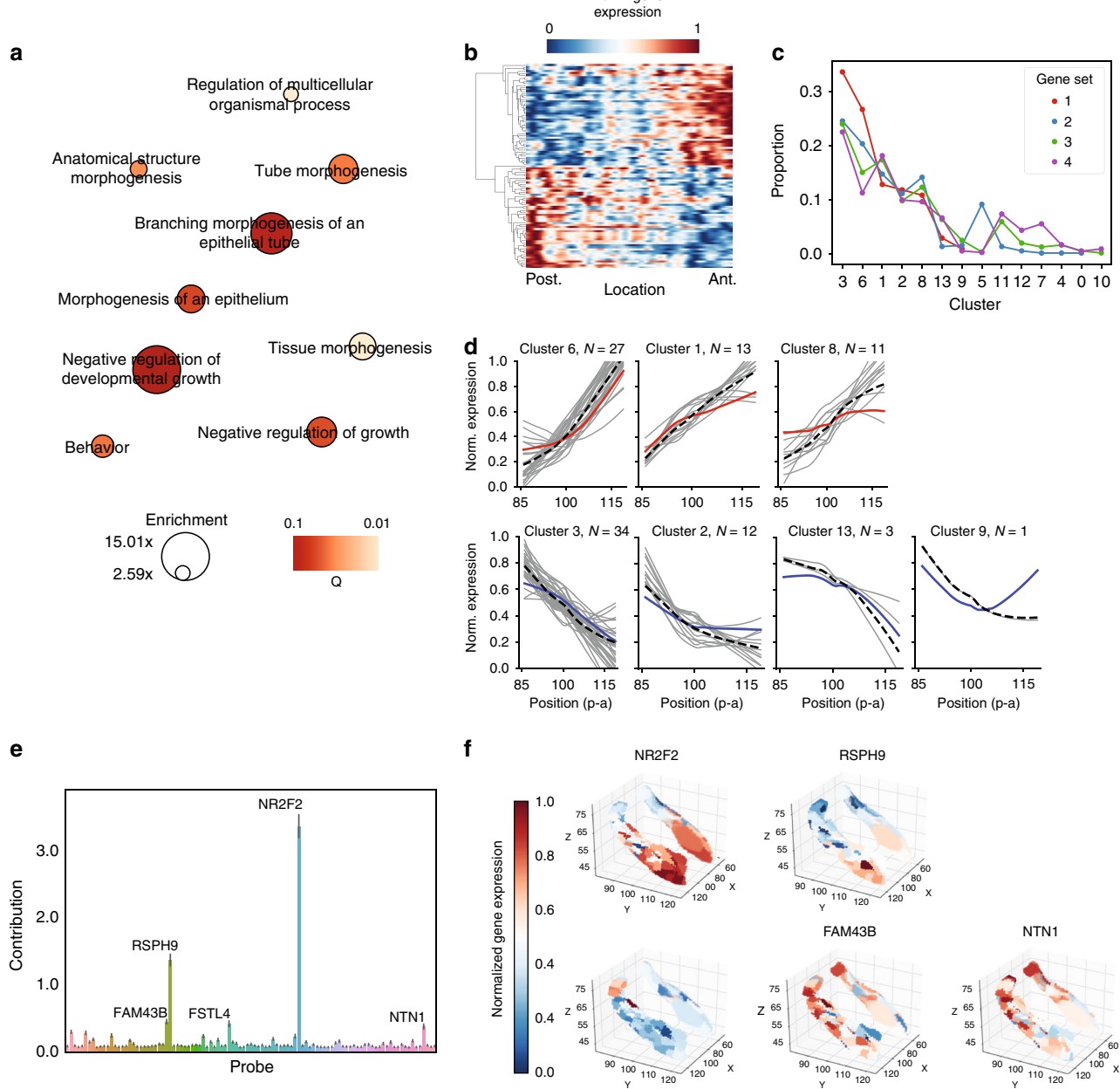

**Fig. 2 Candidate genes associated with the longitudinal axis of the human hippocampus. a** Enriched Gene Ontology terms ($Q < 0.1$) associated with Gene Set 1. Circle size indicates enrichment, whereas color indicates $Q$ value (lighter = lower $Q$ value). **b** Matrix showing gene expression for probes in Gene Set 1 ($y$-axis) across each hippocampal sample, ordered most posterior to most anterior ($x$-axis). Values were smoothed with a 3 mm gaussian kernel across the $x$-dimension only and then clustered so that anterior–posterior patterns can be clearly visualized. **c** Relative proportion of features belonging to each expression pattern cluster, for each Gene Set. **d** Each subpanel represents an expression pattern cluster, and the subpanel heading includes the number of probes from Set 1 assigned to that cluster. For each cluster, the posterior-anterior normalized expression pattern is shown for each Gene Set 1 feature belonging to that cluster (gray), the mean of Set 1 features belonging to that cluster (black dashed), and the mean of features across all sets belonging to that cluster (colored). **e** Average absolute local feature importances (and by extension, model contribution) of probes in Gene Set 1 measured using a Random Forest-based feature explainer across all samples. Error bars = standard error of mean. **f** Surface rendering of the expression patterns of each of the five genes identified as locally important features to predicting position along the longitudinal axis.

found in Set 3. Interestingly, of all probes in the model, NR2F2 and RSPH9 had the highest absolute beta estimates (weights), once again implicating these two genes as potentially important in characterizing the longitudinal axis of the human hippocampus. In addition, to gauge the value of these algorithms over a more simple approach, we ranked all features by their correlation with position along the longitudinal axis, and identified the top 50 positive and top 50 negative correlated features. Twenty-one of

these features overlapped with Set 1 from our previous analysis (see Supplementary Fig. 4d, Data 1 for genes overlapping with Set 1 and the top 100 PLSR features). However, when building a LASSO-based model (as with Fig. 1h) or a simple linear regression model out of these sets, cross-validation accuracy was substantially higher for Set 1 than the PLSR or top-ranked correlation sets (Fig. 4e, f), suggesting the LASSO-PCR model we use does provide some benefit over more simpler approaches.

**Cell-type variation along the long axis of the hippocampus.** Many of the genes identified as top features varying along the longitudinal axis of the hippocampus have also been highlighted as markers for specific genes (e.g. refs. [22–24]). We therefore sought to understand whether the variation in gene expression we observed might reflect variation in cell type along the longitudinal axis. We used two different pipelines designed to infer cell type fraction[25] or expression[26] based on bulk microarray gene expression, each using a different reference dataset for cell types.

Using CiberSortX[25], a reference dataset of cells provided by the Allen Brain Atlas, and cell types identified by Hodge et al.[22], we found 15 cell types with consistent expression within our hippocampus samples (Fig. 5a). All fifteen cell types showed differential expression across subfields (Supplementary Data 5). Three cell types were associated (FDR < 0.1) with longitudinal axis position: mature astrocytes, Tyrobp-positive microglia and a subtype of excitatory neurons (Supplementary Fig. 5b, c, Supplementary Data 5). All three cell types were estimated to be more prevalent in anterior hippocampal areas.

Using a completely different pipeline[26] and a different reference dataset of 35 cell types provided by Lake et al.[24], we found six cell types associated (FDR < 0.1) with axis position (Table S5): four excitatory neuronal subtypes and astrocytes expressed more prevalent in the anterior hippocampus, and one excitatory neuronal type expressed more in the posterior hippocampus (Supplementary Fig. 5b, d). Based on comparisons between the Lake et al.[24] and Hodge et al. cell types[22], the excitatory cell type identified to vary over the longitudinal axis in the CiberSortX analysis was not reproduced in this second validation analysis. Therefore, the only cell type to be reproduced across both approaches was the astrocyte type (Supplementary Fig. 5b).

**Genetic signature predicts sample location in prenatal hippocampi.** We next assessed whether the gene sets identified in our sample of adult human hippocampi could predict the location of samples extracted from the hippocampi of a separate dataset of deceased prenatal humans, aged 15–21 post-conception weeks[27]. Stereotaxic coordinates for these neonatal tissue samples were not available, but precise anatomical labels indicated whether samples were extracted from the caudal or rostral portion of the hippocampus. We applied the models described in the previous sections to predict the location of samples from the neonatal hippocampi. Samples extracted from the rostral hippocampus were predicted to be significantly more anterior than samples extracted from the caudal hippocampus, and this finding was consistent whether the full model was used or models trained on the smaller gene sets were used (Fig. 3). Overall, a slight anterior bias was observed in the predictions. Logistic regression was used to determine whether the pattern of model expression trained in the adult hippocampus could be used to predict whether a sample was extracted from the rostral or caudal portion of the prenatal hippocampus. The model using all genes predicted the location of samples with a 77% accuracy (AUC = 0.86), while prediction accuracy of models using the smaller gene sets ranged from 66% (Set 1) to 83% (Set 2). ROC curves are visualized in Fig. 3.

**The hippocampal long axis signature is echoed across the brain.** The Allen Human Brain Atlas data comprises 3702 samples across the brains of six donors. By leveraging the weights of our LASSO-PCR model, we created the Hippocampal Axis Genomic Gradient Index of Similarity (HAGGIS), a value representing the degree to which the genomic signature of the hippocampal longitudinal axis is represented in the gene expression profile of a given non-hippocampus sample (Supplementary Fig. 1). Larger positive values represent greater genomic similarity to the anterior

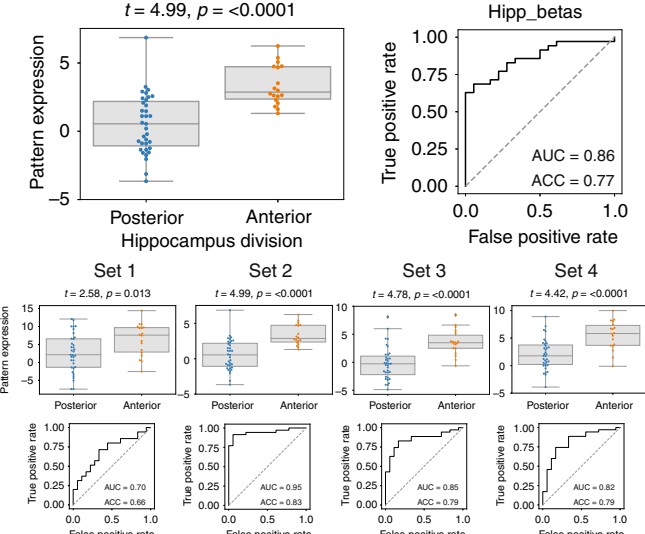

**Fig. 3 Model trained on adult hippocampus predicts location of samples from prenatal hippocampus.** The pattern of gene expression used to predict the location of adult hippocampus samples along the longitudinal axis were applied to samples extracted from the prenatal human hippocampus. Boxplots compare this normalized pattern of expression between samples extracted from the rostral and caudal hippocampus. Statistics are calculated using t-tests. ROC curves were generated using logistic regression. The top figures correspond to models fit using all features, whereas the bottom figures correspond to models fit using the smaller gene sets. ACC Accuracy. AUC Area Under the Curve. For boxplots, the center line, boxes and whiskers represent the median, inner quartiles, and rest of the data distribution (except outliers), respectively.

hippocampus, while smaller negative values represent greater genomic similarity to the posterior hippocampus. When plotting these values for all brain samples, we observed a general pattern across the brain such that the brainstem and more antero-ventral sites of the cerebral cortex demonstrated greater genomic similarity to the anterior hippocampus, whereas the cerebellum and posterio–dorsal cortical regions demonstrated greater similarity to the posterior hippocampus (Figs. 4 and 5a).

**Genomic interactions with dissociated hippocampo–cortical systems.** The anterior and posterior hippocampus each exhibit a distinct profile of anatomical connections in humans[9], which can also be represented using resting-state functional connectivity[6]. Using logistic regression and the HAGGIS, we identified coordinates to isolate the genomic posterior and anterior hippocampus (Supplementary Fig. 7a). We then used an open database of resting-state functional connectivity information based on rsfMRI scans from 1000 subjects to create an average voxelwise map representing the degree to which brain regions are functionally connected to the anterior vs. posterior hippocampus. Brain samples bearing a gene expression profile more similar to the anterior hippocampus were also more functionally connected to this substructure, while the opposite pattern was observed for samples with gene expression profiles more similar to the posterior hippocampus ($r^2 = 0.170$, Fig. 5a). Disagreement between these two cortical patterns was mostly a function of magnitude, but some major directionality differences were observable in salience network and posterior default mode network regions (Supplementary Fig. 6). A separate model was constructed in order to ascertain the maximum (cross-validated) variance in differential connectivity explainable given the (genomic) data. This analysis revealed that, while HAGGIS explained only 17% of

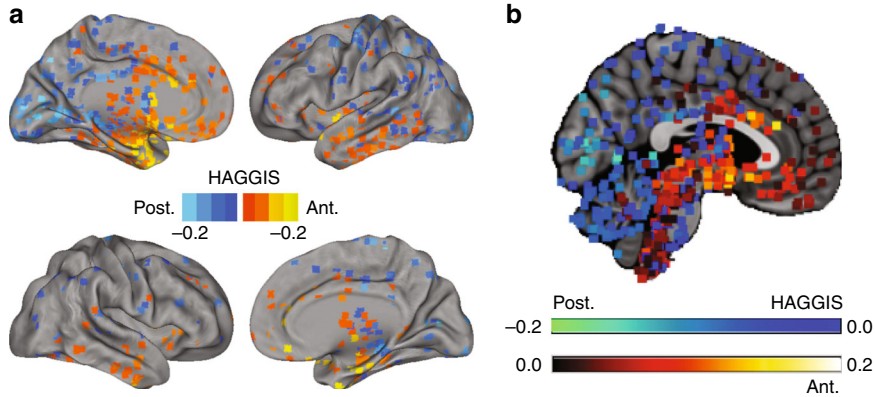

**Fig. 4 Spatial distribution of the HAGGIS across the brain. a** Each sample was projected onto a cortical surface based on its MNI coordinates. Warm colors indicate the sample has a gene expression pattern more similar to the anterior hippocampus (higher HAGGIS), while cool colors represent the sample is more genomically similar to the posterior hippocampus (lower HAGGIS). **b** A medial slice inclusive of brainstem and cerebellum. Each dot represents a sample, and warm colors indicate higher HAGGIS, while cool colors represent lower HAGGIS. HAGGIS Hippocampal Axis Genomic Gradient Index of Similarity.

the total model variance, it explained about 51% of the variance explainable with the present genomic data (Fig. 5c).

The strength of this relationship differed depending on where along the anterior–posterior axis the divisions were drawn, which parts of the brain were included, and the size of the cube used to extract data around the sample coordinate (Supplementary Fig. 7c). The $r^2$ ranged from 0.111 (central split, cortical only mask, 1 mm cube diameter) to 0.304 (split at anterior/poster extremes, mask excluding only brainstem and cerebellum, 11 mm cube diameter), though in all cases the relationship was observed to be significantly greater than chance (95% CI of chance $r^2 <$ 0.004 for all conditions). The relationship between HAGGIS and functional connectivity also varied slightly depending on the gene Set used (Fig. 5b). Remarkably, prediction of functional connectivity by HAGGIS performed just as well when the HAGGIS was created using the smaller Sets, with the highest values achieved when only the top 100 probes were used.

A diverging pattern of structural covariance with the rest of the brain has also been observed across the longitudinal axis of the hippocampus[28], perhaps representing co-variation in cytoarchitecture. We used an open dataset of 153 structural MRI images from young healthy individuals to create a map representing variation in structural covariance between the brain and the anterior vs. posterior hippocampus. The more similar a brain region's gene expression patterns were to the anterior hippocampus, the greater the structural covariance was between that structure and the anterior hippocampus, and vice versa for the posterior hippocampus ($r^2 = 0.284$; Fig. 5a). HAGGIS explained 62% of the variance explainable with the present genomic data Fig. 5c). This relationship varied but remained strong across different brain masks and gene sets (Fig. 5b). Some disagreement in directionality between these two cortical patterns was observed in the anterior and posterior cingulate, anterior insula, and medial occipital cortex (Supplementary Fig. 6).

To validate these findings without relying on an anterior–posterior split, we utilized a previously validated data-driven approach[6,29,30] to extract the principal gradients of hippocampal functional connectivity and structural covariance with the rest of the brain, respectively. We then tested the relationship between each gradient and the predicted location of each sample based on the HAGGIS (Supplementary Data 6). For structural covariance, the 1st gradient, explaining 24% of the total variance in brain-hippocampus covariance, showed a strong correlation with HAGGIS ($r^2 = 0.41$; Supplementary Fig. 7d). For functional connectivity, the 3rd gradient, explaining 13.5% of the

total variance of hippocampus–brain connectivity, also showed a strong relationships with HAGGIS ($r^2 = 0.40$; Supplementary Fig. 7e). These findings were not contingent on the gene set used to calculate the HAGGIS (Supplementary Fig. 7f).

**Genomic associations with regional neurodegenerative vulnerability.** The anterior and posterior hippocampus are also differentially involved in disparate neurodegenerative diseases[18], particularly Alzheimer's disease (AD) and frontotemporal dementia (FTD)[10–12]. We acquired fluorodeoxyglucose (FDG) PET scans measuring glucose metabolism, a measure of neuronal health and degeneration, from patients diagnosed in a tertiary memory clinic as having AD or FTD. We used these scans to create a statistical map representing the relative patterns of neurodegeneration in AD vs. FTD. We found that samples with greater genomic similarity to the anterior hippocampus also showed greater hypometabolism in FTD compared to AD, whereas samples more similar to the posterior hippocampus showed greater hypometabolism in AD compared to FTD ($r^2 = 0.118$; Fig. 5a). Disagreement between these cortical patterns was mostly relegated to magnitude differences in the temporoparietal cortex, though some directionality differences were observed in the motor cortex (Supplementary Fig. 6). HAGGIS explained about 21% of the variance explainable given the present genomic information (Fig. 5c). This relationship also varied depending on the regions included and cube size, with $r^2$ ranging from 0.095 (whole-brain, 1 mm cube diameter) to 0.153 (cortex-only mask, 11 mm cube diameter, Supplementary Fig. 9), but remained greater than chance in all cases. Notably, and unlike previous analyses, the relationship between HAGGIS and regional disease vulnerability was not observed when restricting the HAGGIS to the top 100 probes (Set 1) (Fig. 5b). A post hoc analysis of Set 2 and Set 3 gene ontology clusters determined in the section (Genes associated with the long axis of the human hippocampus (in Results)) revealed that genes associated with cell motility and axon guidance (enriched terms from Set 1) were not strongly associated with disease vulnerability. The association with HAGGIS and disease vulnerability in Sets 2 and 3 were instead driven by genes associated with amine activity, phosphorylation, hormonal signaling, serotonin binding and vascular growth factor activity (Supplementary Fig. 10).

**Specific genes link long axis to connectivity and vulnerability.** In order to highlight specific genes that may be involved in both

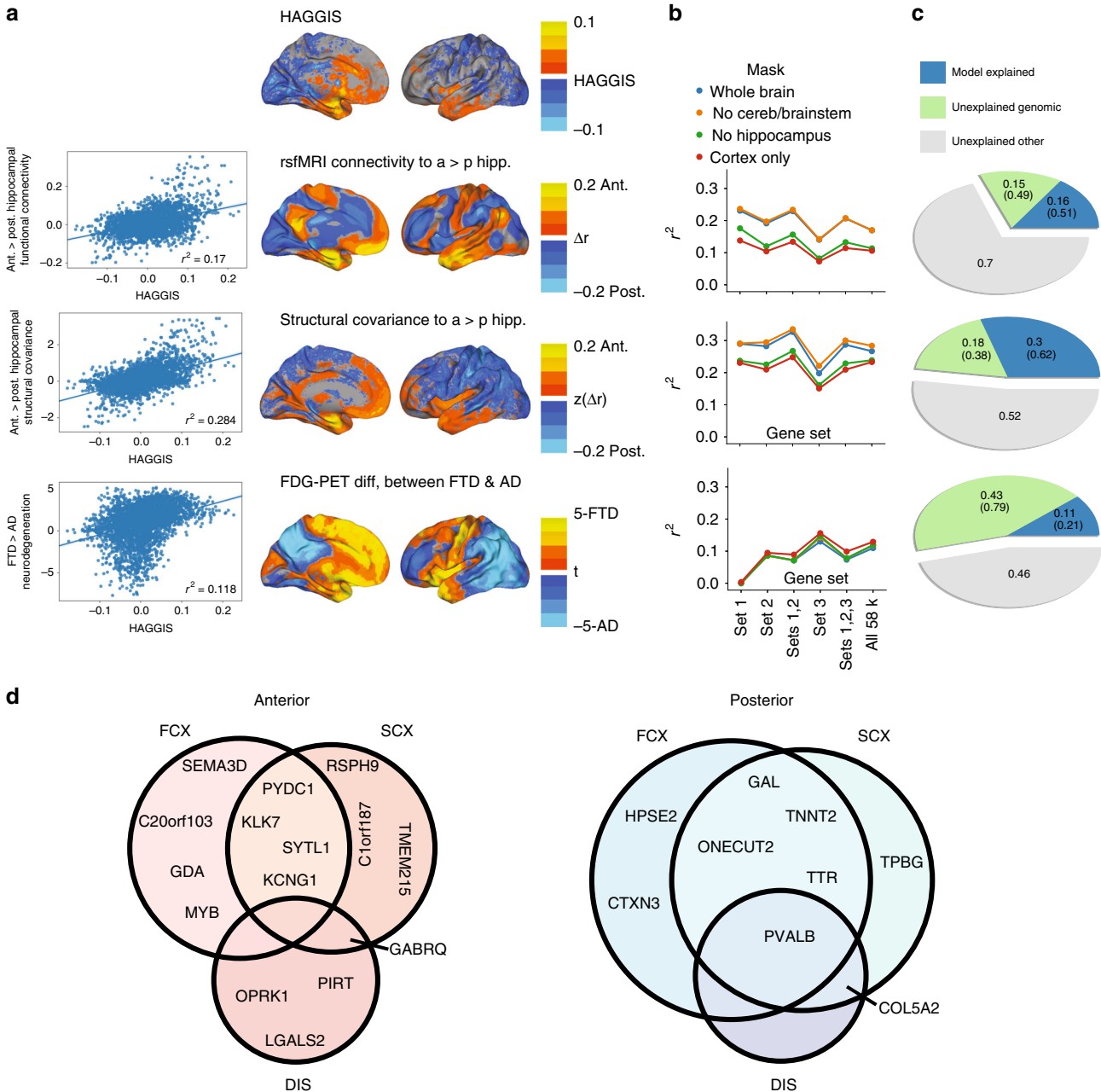

**Fig. 5 HAGGIS predicts hippocampus–brain relationships. a** From top to bottom: The spatial distribution of (smoothed) HAGGIS across samples, differential functional connectivity to the anterior vs. posterior hippocampus measured with rsfMRI, differential structural covariance with the anterior vs. posterior hippocampus, differential vulnerability to AD or FTD measured with FDG-PET. Graphs on the left visualize the relationship between these spatial patterns by comparing the HAGGIS of each sample with the mean value from the respective map within a 5-voxel cube around the sample coordinate. **b** Each of the above associations was re-calculated using three other brain masks, and using a HAGGIS formed from each gene set identified in the section (genes associated with the long axis of the human hippocampus in Results). The $r^2$ of each of these associations is visualized. **c** Pie charts indicating the proportion of genomic and total variance explained by each model. Numbers in parentheses indicate percentage of total genomic variance. **d** Genes involved in both the longitudinal axis of the hippocampus, and hippocampus–brain interactions. All genes pictured are among the top 50 anterior (red; left) or posterior (blue; right) features of the hippocampus longitudinal axis model. Each also participates in one or more hippocampus–brain interactions, indicated by the circles within the Venn diagrams. FCX Differential functional connectivity between anterior and posterior hippocampus; SCX Differential structural covariance between anterior and posterior hippocampus; DIS Differential vulnerability between AD and FTD.

maintenance of the longitudinal axis and hippocampus–brain interaction, we constructed independent models to learn the genomic profile of the maps from Fig. 5a and compared the top 100 features from these models to the longitudinal axis model. The proportion of overlap between the top 100 features of each model with the top 100 features from the hippocampus longitudinal axis model far exceeded chance (functional: 20%;

structural: 21%; disease: 11%). Overlapping genes from each model, stratified by involvement in anterior or posterior hippocampus, can be found in Fig. 5d. Interestingly, some genes were involved in multiple systems. For example, PVALB, specifically expressed in the posterior hippocampus, was also highly expressed in brain regions functionally connected and structurally covarying with the posterior hippocampus, as well as in regions

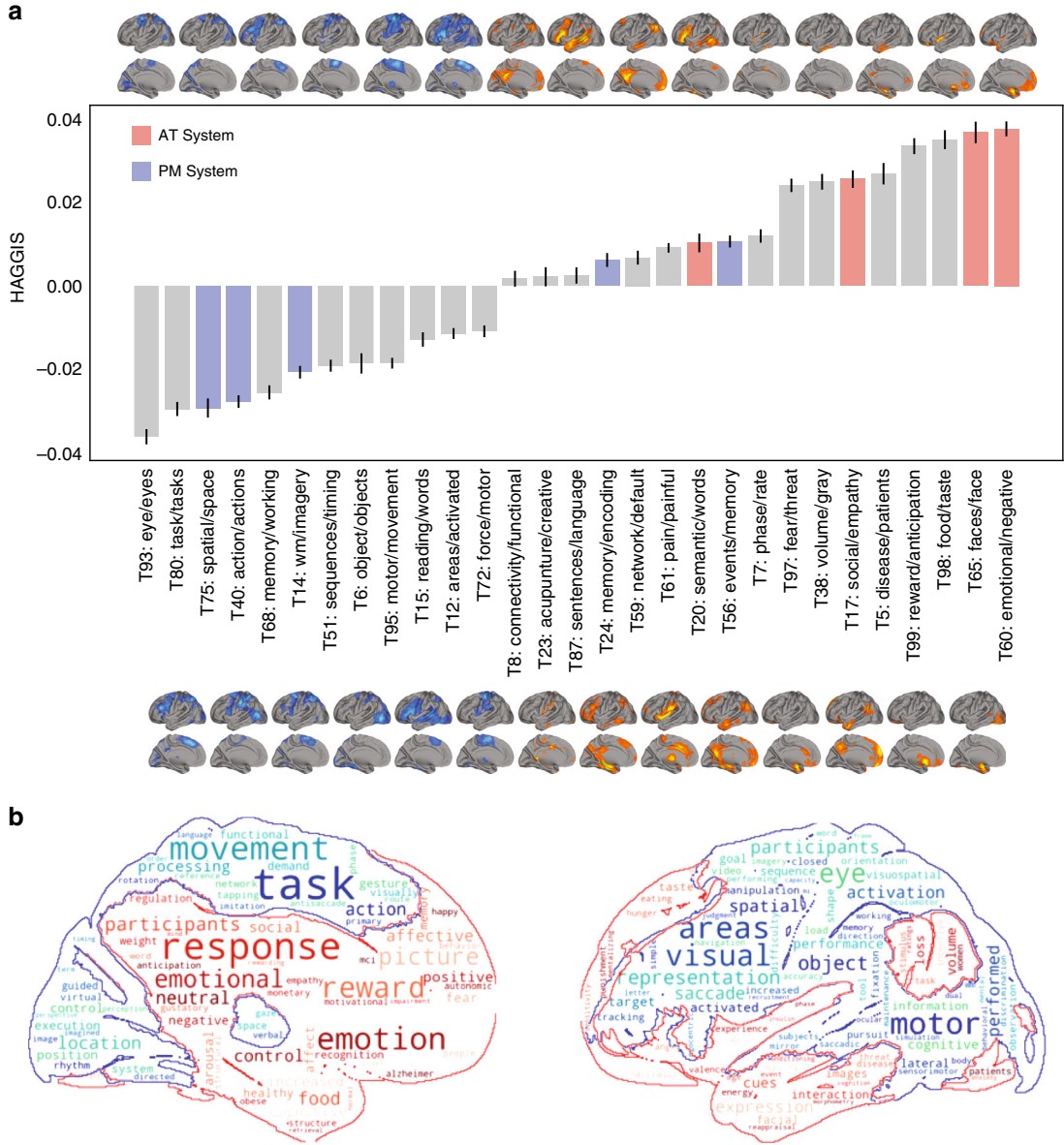

**Fig. 6 Variation in genomic signature predicts involvement in distributed cognitive networks. a** Maps were downloaded from neurosynth representing greater than chance meta-analytic functional activation in studies with different topic-sets mentioned in their text. Mean HAGGIS (represented by bars) was calculated for samples inside maps encompassing >500 samples (visualized either directly above or directly below each bar). Error bars represent standard deviation. Topics hypothesized to belong to the AT or PM system are shown in red and blue respectively. **b** A word cloud summarizing the regions and topics most associated with the genomic signal of the anterior (red) and posterior (blue) hippocampus. Larger words are more associated with networks with higher (red) or lower (blue) HAGGIS.

specifically vulnerable to Alzheimer's disease. Additionally, anterior hippocampus gene GABRQ was also highly expressed in regions both structurally covarying with the anterior hippocampus and those vulnerable to frontotemporal dementia.

**Long axis signature and involvement in cognitive networks.** The posterior and anterior hippocampus are implicated in distinct aspects of memory and cognition[6–8,31]. We explored whether regions sharing genomic similarity to the posterior or anterior hippocampus were more likely to participate in cognitive networks proposed to involve those substructures. We downloaded 100 meta-analytic functional coactivation maps from the Neurosynth database, each composed from between 91 and 4201 task-based functional MRI activation studies, and each of which was paired with a set of related cognitive/behavioral topic sets.

These topic/map pairs represent greater-than-chance regional functional coactivation patterns reported consistently in studies sharing words from certain related topic-sets in the publication text. These maps can therefore be thought to represent specific region-sets involved in distributed cognitive networks. We calculated the mean HAGGIS of samples falling within each cognitive map, with higher positive values indicating greater genomic covariance between the regions covered by that coactivation map and the anterior hippocampus, and lower negative values representing greater genomic covariance of those regions with the posterior hippocampus.

Using a conservative approach (only including maps with at least 500 overlapping tissue samples: 29 maps; minimum map size: 36,622 voxels), we observed a pattern largely consistent with previous hypotheses of hippocampal involvement in different cognitive systems[10] (Fig. 6). As we hypothesized, regions that

expressed a gene expression profile more consistent with the anterior hippocampus tended to be those involved in social and emotional cognition, but also included maps associated with reward and conditioning, among others. Also consistent with our hypotheses, cognitive networks more genomically similar to the posterior hippocampus were associated with spatial cognition, imagination and mental simulation, but also included maps associated with visualization, working memory and movement/action. Interestingly, maps associated with episodic memory and physical stimulation slightly favored the anterior hippocampus, or were not strongly associated with either posterior or anterior hippocampus. These patterns remained remarkably similar when repeating the analysis with only probes from Set 1, representing the top 100 probes in our model (Supplementary Fig. 8).

## Discussion

The hippocampus plays a central role in many systems that regulate behavioral processes across several species, and that are dysregulated in several human neuropsychiatric diseases. The contribution of the hippocampus to many of these systems is grossly organized across its longitudinal axis. Characterizing the molecular properties of this axis may be vital to understanding how gene expression networks regulate macroscopic brain networks. We show that the anterior–posterior position of a tissue sample extracted from the adult human hippocampus can be predicted with remarkable accuracy using the expression pattern of a small set of genes, and that this pattern generalizes to the developing hippocampus of prenatal humans. Further, we find genomic representation of the anterior–posterior gradient projected across the entire brain, and that this representation partially explains relationships between the hippocampus and dissociated hippocampo–cortical systems. The anterior hippocampus shares genomic patterning with a system encompassing the medial prefrontal cortex, anterior temporal lobe and the brainstem. In general, these regions showed greater functional connectivity and structural covariance with the anterior than the posterior hippocampus, greater vulnerability to FTD than to AD, and more frequent participation in cognitive tasks involving motivation/conditioning, social and emotional cognition and semantic knowledge. The posterior hippocampus, in contrast, shared a genomic pattern with the cerebellum, and occipital, temporoparietal and motor and pre-motor cortex. These regions generally showed greater connectivity and structural covariance with the posterior than anterior hippocampus, more vulnerability to AD than FTD, and were more likely to participate in cognitive tasks involving spatial representation, visual processing, working memory and simulation. These results confirm and extend findings across species and sub-disciplines of neuroscience to suggest shared gene expression patterns underlying a well-described dissociation of anterior vs. posterior hippocampal involvement in cognitive brain networks. Further, the findings support the existence of a specific axis of organization in the human brain, where an anterior–ventral–posterior–dorsal gradient explains regional involvement in diverse behaviors, underscored by a specific pattern of gene expression (Supplementary Fig. 11). These findings together form a template for studying how specific genes may regulate the development of dissociated hippocampal connectivity networks in humans and their involvement in specific behaviors and, potentially, specific diseases.

Our results support an existing concept of molecular gradients in the cerebral cortex[29,32,33]. The anterior–ventral–posterior–dorsal pattern observed in our data is reminiscent of a general anterior–posterior molecular gradient previously observed in the Allen Human Brain Atlas dataset[33,34]. Fornito et al.[33] reviews qualities of this gradient, including a pattern of neuronal organization where, as one moves caudally to rostrally, neuron and arbor

size increase while neuronal number and density decrease. Perhaps related, a dual origin hypothesis of diverging cortical systems has been proposed suggesting the cerebral cortex has developed radially from certain phylogenetically conserved limbic structures over the course of evolution. This hypothesis describes a ventral system emanating from the perirhinal and amygdalar cortex that is involved in semantic identification of a stimulus and motivated behavior, while a dorsal system has evolved from the hippocampus and parahippocampal cortex to coordinate spatial representation and coordinated action[35]. The hypothesis is supported by evidence from comparative cytoarchitectonics and connectivity patterns across species. The anterior temporal/posterior medial (AT/PM) hypothesis of memory systems[10] provides yet another example of opposing cortical systems loosely following an anterior–posterior organization and determining patterns of brain organization. Each of these three models originated from a different field of inquiry—gene expression, cortical evolution, memory network organization—but the models converge in many respects, and a microcosm of this shared framework seems to be represented along the longitudinal axis of the hippocampus—explicitly so in the AT/PM and dual origin models. Our results generally support the premise that the hippocampus participates in two distinct macroscopic networks characterized by distinct structural covariance, functional connectivity, behavioral domain specificity and disease vulnerability, and that participation in these networks can be predicted by position along the longitudinal axis. We take this framework one step further to suggest these two distinct networks are composed of one single gradient of gene expression that could play an upstream role in their systemic distinctions.

The genomic gradient we have identified may be pertinent to the origins of some of the cortical specificity described above, however there are also a number of discrepancies worth note. For example, functional connectivity and structural covariance patterns between the hippocampus and salience network and posterior default mode network regions were inconsistent with what would be expected given genomic similarity to the hippocampus (Supplementary Fig. 6). Related, while the posterior DMN showed greater genomic similarity to the posterior hippocampus—as would be predicted by the AT/PM system[10]—the medial prefrontal cortex was much more associated with the anterior hippocampus. In addition, regions meta-analytically active during episodic and autobiographical memory and encoding/retrieval tasks showed a genomic profile more similar to the anterior hippocampus. However, in our own resting-state fMRI analysis utilizing information from 1000 individuals, DMN structures were more connected to the anterior than the posterior hippocampus. These findings coincide with results from a previous large-scale meta-analytic coactivation study of differentiated hippocampal function along its longitudinal axis[8], and are consistent with a recent study observing changes in functional connectivity across the longitudinal axis of the human hippocampus[6].

Regarding other discrepancies, we should note that neither the Dual Origin hypothesis nor the AT/PM model include the cerebellum or brainstem, structures showing prominent divergence along the hippocampal gradient in question. This divergence may suggest that the molecular gradients defining anterior–posterior divergence in the cortex define similar divergence in subcortical structures, and point perhaps to dorsal and ventral plate patterning during neural development[36]. While we have observed much agreement between HAGGIS and a number of other gradient-like neural organization patterns, the disagreements (summarized in Supplementary Figs. 6 and 11) leave much to be elucidated about how developmental and environmental molecular signals contribute to the convergence and divergence of these various systems.

While our initial model utilized the expression patterns of nearly 60,000 probes corresponding to over 20,000 genes, the model favored a much smaller profile of probes to describe the longitudinal axis of the human hippocampus. When isolating a small set of only 100 probes, we were able to successfully predict the location of samples along the longitudinal atlas with <2 mm error, as well as interactions between the hippocampus and the specific brain systems. The set was enriched with genes associated with, in particular, anatomical structure morphogenesis and cellular growth, suggesting genes within this set may be involved in coordinating and/or maintaining the anatomical variation of the hippocampus along its longitudinal axis. Whether these genes are also partially responsible for the functional variation along the axis remains unclear, though it is notable that similar expression patterns of these 100 genes can be observed in other brain regions that interact with the hippocampus. In particular, we identified several specific genes that appear to be involved in coordinating both the longitudinal axis of the hippocampus and one or more aspects of the hippocampus-associated distributed brain networks. A number of these genes (PVALB, GAL, ONECUT2, PIRT, TNNT2, RSPH9, COL5A2, CTXN3) have been reported in previous studies examining genes associated with functional network organization[37,38]. Our work also converges with other studies showing gene expression across disparate regions may be related to shared anatomical characteristics (e.g. refs. [39–41]). Closer examination of the contribution of genes across these different network properties (structure, function, disease vulnerability) is warranted, and may lead to a clearer picture of the role various microscale features play in macroscale network organization.

Many of the genes identified in our study have also been described in previous studies characterizing the dorsal and ventral subdivisions and longitudinal gradients of the rodent hippocampus (e.g. refs. [13,14,16,18,19], see Table 1). This suggests a fair degree of homology between rodents and human in the distribution of proteins along the longitudinal axis of the hippocampus, and perhaps in the development and maintenance of the axis itself. However, many previously undocumented proteins were also identified, and replication and comparative studies will be required to disentangle whether these candidate genes are truly unique to humans or a result of small sample sizes and differing methodologies.

The most important genes identified in our model can be interpreted as the those genes that were central to the specific hippocampal gene expression network(s) that were most strongly associated with position along the longitudinal axis of the hippocampus. We cannot infer which genes are causally related to axis formation and maintenance and, as weights from backward regression model are notoriously hard to interpret[21], even identifying the most important among a set of genes is challenging. Being aware of these limitations, we identified NR2F2 (also called COUP-TFII) and RSPH9 to be particularly important in local prediction of sample location along the axis. This likely suggests that these two genes demonstrated the cleanest and most consistent linear gradient in expression across the longitudinal axis among those assessed (which can be visually appreciated by the surface plots of expression levels of these genes across the hippocampus in Fig. 2). The pattern of expression we observed here mirrors patterns from other studies describing NR2F2 expression in the rodent hippocampus[42], as well as more macroscopically in the human brain during development, particularly in the temporal lobes[43]. NR2F2 is also key in the determination of cell fate in numerous circumstances[44,45], including that of interneurons expressing PVALB (parvalbumin) or SST (somatostatin), where NR2F2 promotes SST and represses PVALB[45]. These findings are highly consistent with the expression of PVALB (expressed posteriorly) and SSTR1 (expressed anteriorly with NR2F2) in our data (Table 1), of particular interest given the role of these two

genes in marking distinct and anticorrelated structural and behavioral brain systems[46–48]. For its part, RSPH9 is part of the structure of primary cilia, which can be found within ependymal cells lining the ventricles, as well as in the CA1 subfield of the hippocampus and adjacent choroid plexus[23]. There is evidence that these cilia can promote neurogenesis in the hippocampus through mediating expression of SHH (sonic hedgehog)[49], a protein implicated in anterior–posterior pattern formation, and identified as a possible protein of importance in our data by the presence of HHIP (hedgehog-interacting protein) among the top 100 anterior–posterior associated genes (Table 1). The expression patterns of RSPH9 in our data may signal the presence of a specific pattern of cilia, which may help regulate the longitudinal axis of the hippocampus through hedgehog signaling. Both NR2F2 and RSPH9 have been identified as role-players in human functional network organization[37,38].

Several other genes of interest appeared among are top 100 features, some of which are associated with specific cell types and cell classes (e.g. PVALB, GABRQ, TMEM215, SST for interneurons)[22–24]. However, we did not find strong evidence for cell type variation across the longitudinal axis of the hippocampus, with the exception of a trend of increase of tufted astrocytes[22] toward more anterior sites of the hippocampus. This latter finding supports histological data specifically in the murine dentate gyrus[50]. Generally, our results instead suggest variation of gene expression within cells along the long axis, rather than variation in the types of cells themselves, though these findings remain preliminary due to methodological limitations.

Data from multiple studies suggest a specific role for the longitudinal axis of the hippocampus in AD and FTD[11,12]. Our data support this notion, suggesting that regions more vulnerable to FTD than AD share a more similar molecular profile to the anterior than posterior hippocampus, and that the opposite pattern was observed for regions more vulnerable to AD than FTD. This relationship is far from perfect—for example, the cerebellum is not thought of as a vulnerable region in AD, nor is the medial occipital cortex (though our results showed the occipital cortex is empirically more impaired in AD than in FTD). Similarly, while neurodegeneration in anterior and middle temporal lobes are more severe in FTD, these regions are also vulnerable in AD. However, our results contrasting these two neurodegenerative diseases directly highlights a general divergence in vulnerability across the anterior–posterior axis of the brain, mirroring the extreme of the HAGGIS gradient. While it is tempting to wonder whether the same genes that coordinate the development of different systems also incidentally contribute to the degeneration of these systems over time, post hoc analyses suggest that factors associating with disease vulnerability may be downstream from those factors associated with anatomical brain development (Supplementary Fig. 10). Although little can be extrapolated from our data about the potentially dissociated role of specific proteins in AD and FTD, we provide evidence for distinct molecular properties that characterize the dissociated hippocampo–cortical systems vulnerable to each of these two diseases. The implicated genes and proteins may provide promising candidates for more targeted studies of disease-specific neurodegeneration.

Our study comes with a number of important limitations that must be addressed. The single greatest limitation of our study is that our gene expression data comes from a limited number of samples taken from only six donors who differed in age, sex, and ethnicity. We partially addressed this issue by regressing out donor effects from our gene expression data and performing leave-one-donor-out analyses, but in doing so, assume certain aspects of gene expression should be fairly consistent across individuals. Some confidence is inspired by the fact that, in spite of these limitations, we were able to replicate findings from

rodent studies. We also tried to circumvent this issue by showing that relationships linked to our primary findings hold in several other independent datasets. Another major limitation is a reliance on specific coordinates of samples reported at time of autopsy, translated to single-subject MRI space, and then normalized to a common subject space. While we took measures to improve the quality of the normalization to common space, we cannot rule out noise introduced during any of these steps. Our analysis of gene expression gradients along the hippocampal longitudinal axis is particularly sensitive to these issues because it relies on the exact coordinates of the samples extracted. Once again, we were able to replicate findings from other studies, but it is possible that the importance of some gene signatures to our model could have been affected. As discussed previously, another limitation is related to our attempts to extrapolate biological importance from machine learning models. While we took many steps to try to test the stability of weights in our models, our interpretations remain somewhat speculative and must be replicated in more focused studies. However, we are encouraged by the presence of several genes identified in previous studies of the rodent hippocampal long axis, and by generalization of our model to an external dataset. The fact that we are using microarray expression data is also an important point worth mentioning—we are measuring transcripts from genes but we do not know to what extent these transcripts result in protein expression. With regard to interpretations, an important point to note is that, while only 100 genes were sufficient for statistical characterization of the hippocampal long axis, nature likely requires coordination among many more genes. Finally, a major limitation comes with the complexity of drawing conclusions across many datasets, each of which are subject to variation based on methodological processing. We tried to overcome this by primarily using open-access data preprocessed beforehand by experts, and by making all of our data and code freely available at https://github.com/illdopejake/Hippocampus_AP_Axis[51] so that other researchers can scrutinize, reproduce, and hopefully re-use our analyses.

## Methods

**Human gene expression data.** Human gene expression data were downloaded from the Allen Human Brain Atlas (http://human.brain-map.org, RRID: SCR_007416). A detailed description of this dataset can be found elsewhere[20,52,53]. Briefly, tissue samples were extracted across both hemispheres of two human brain donors, as well as the left hemisphere of four additional donors, totaling 3702 samples. Metadata for the six donors can be found in Supplementary Data 7. Stereotaxic coordinates and MNI space coordinates are provided for each sample. Each sample underwent microarray analysis and preprocessing to quantify gene expression across 58,692 probes. This analysis provides an estimate of the relative expression of different transcripts (encoded by different genes) within the tissue sample. While previous publications have used different strategies to reduce the number of probes (see ref. [53] for review), due to assumptions associated with these strategies and the high-dimensionality approach of our models, we opted to retain all 58,692 probes for analysis.

Importantly, the MNI coordinates originally supplied with the dataset did not account for nonlinear deformations in transforming the donor MRIs in native space to MNI space, and thus included a noticeable degree of error (i.e. many samples mapped outside of the brain or their labeled brain regions)[53]. However, these coordinates have been meticulously reconstructed and transformed accounting for nonlinear deformations[54]. Moving forward, all mentions of MNI coordinates will refer to these Devenyi coordinates.

Given the different ages, sexes and other characteristics, substantial differences in gene expression are expected between donors. However, similar to previous studies using this dataset, we were only interested in common patterns of human gene expression for the present analyses, rather than inter-individual differences. As such, all samples across the six donors were aggregated and we regressed donor-specific effects from each probe using linear models. Specifically, we used dummy coded donor ID variables to model donor-specific patterns for each probe, and by taking the standarized residuals of this model, removed variance specifically associated to each donor from each probe. Therefore, probe values represent gene expression normalized across all samples, adjusted for inter-individual statistical differences.

Along with coordinates, each sample contains ground-truth information about the specific brain sub-structure from which the sample originated, as defined by the anatomist extracting the sample. To identify samples falling within the

hippocampus, we selected all samples with structure labels of CA1 field, CA2 field, CA3 field, CA4 field, Subiculum and Dentate Gyrus, from both the left and right hemispheres—188 samples in total. In all, 18 samples had MNI coordinates more than 3 mm outside of the hippocampal volume defined below, leaving 170 hippocampal samples in total.

**Identifying the longitudinal axis of the hippocampus.** Many previous studies have explored differences between the dorsal and ventral (or posterior and anterior) hippocampus, but such a system requires an often arbitrary delineation between these two structures[4,5]. To overcome this limitation, we instead sought to quantify the longitudinal axis of the hippocampus and observe changes in gene expression across this axis. Such an approach would still capture gross differences in expression between anterior and posterior sites, but would also allow for detection of more complex gradients. Notably, the hippocampus curves dorsally and medially, so a straight line may not be appropriate for defining its longitudinal structure.

The objective is to identify a curved path that follows the center of mass of the hippocampus along its curvilinear shape (Supplementary Fig. 1b). The initial hippocampus volume was defined as labels 9 and 19 from the Harvard-Oxford-sub-maxprob-thr25-1 mm atlas derived from the MNI ICBM152 average brain template, supplied with FSL 5.0 (RRID:SCR_002823). A "skeleton" of the hippocampal volume was created from morphological operations (dilations/erosions) using the MINC Toolkit (version 1.0.08) (RRID:SCR_014138; http://bic-mni.github.io/#MINC-Tool-Kit). The hippocampus mask was resampled to 0.5 mm isotropic voxel size and a chamfer map was created, measuring the distance from the border of the resampled hippocampus volume up to 10 mm away. This chamfer map was binarized to create a large smooth blob around the hippocampal surface. An opposite chamfer map was created inside the blob, and the local minimum of the derivatives of this map were computed in order to isolate the points at the greatest distance from the blob surface. This creates a "skeleton" following the curvilinear shape of the hippocampal volume, which was then masked with the original hippocampal volume. Finally, the skeleton was resampled back to 1 mm space.

Next, this hippocampal skeleton, in MNI space coordinates, was used to calculate the position of each hippocampus tissue sample along the longitudinal axis. For each sample, we identified the skeleton MNI coordinate with the minimum projected distance to the sample's MNI coordinate. The position of the sample was then coded as the y-coordinate (anterior–posterior axis) of the closest skeleton voxel. This process effectively transforms all sample coordinates along a single anterior–posterior dimension. (Supplementary Fig. 1b). Note that, depending on location of the sample, the MNI y-coordinate of the sample may not share the same y-coordinate of the closest skeleton point.

**Genes associated with the long axis of the human hippocampus.** We sought to identify which specific genes were associated with positioning of samples along the longitudinal hippocampal axis. Sparse regression algorithms built for high dimensional datasets have been proposed, such as least-angle regression (LARS) and LASSO-LARS. However, during regularization, these algorithms will often select only one of a set of several collinear variables and reduce the coefficient of the other variables in the set to zero. In the case of gene expression data, gene co-expression networks are of interest to us, and we do not necessarily want to select one of a set of co-expressed genes. Therefore, we opted instead to use a LASSO-PCR approach[55,56]. Such an approach will reduce the dimensions of the data while preserving gene co-expression networks, yet still allow for a sparse selection of features.

In summary, we reduced our input data, a 170 (sample) × 58,692 (probe) matrix, using principle components analysis (PCA) with singular value decomposition. The resulting 170 (sample) × 170 (component score) matrix was used in a principal component regression (PCR) model (Supplementary Fig. 1). Approaches to PCR models typically reduce the number of independent variables by removing the components whose eigenvalues fall below some threshold related to the percentage of variance explained. This does not account for potentially strong relationships between the dependent variable and minor components. Thus, we elected to use a Least Absolute Shrinkage and Selection Operator (LASSO) regression model with sample position along the longitudinal hippocampus axis (defined in the previous section) as the dependent variable.

In our regression model we have our standardized matrix of gene expression data $\mathbf{X}$, our measurements along the longitudinal axis $\mathbf{Y}$, and the model $\mathbf{Y} = \mathbf{XB} + \epsilon$. We wish to estimate the values of the matrix $\mathbf{B} = [\beta_0, \beta_1, \ldots, \beta_p]^T$, where $\beta_i$ is the estimated impact of probe $i$ on longitudinal position. Probes with larger impacts will have higher estimated values; negative values suggest greater expression in posterior compared to anterior hippocampus, and vice versa.

Since there are a large number of regression parameters, we use dimension reduction through PCA. We transform the data such that $\mathbf{X}^T\mathbf{X} = \mathbf{P}\boldsymbol{\Lambda}\mathbf{P}^T = \mathbf{Z}^T\mathbf{Z}$, where $\boldsymbol{\Lambda}$ is the diagonal matrix of eigenvalues of $\mathbf{X}^T\mathbf{X}$, $\mathbf{Z}$ is the matrix of principal components, and $\mathbf{P}^T\mathbf{P} = \mathbf{I}$. We are now interested in solving the principal component regression $\mathbf{Y} = \mathbf{ZA}$, where the regression coefficients are stored in the matrix $\mathbf{A}$ and are the contribution of principal components to position. We derive estimates of $\mathbf{A}$ using LASSO. The coefficients of the two regression equations are

related by the expressions $A = P^TB$ and $B = PA$, so we estimate $\hat{B} = P\hat{A}$, giving us the beta values of the individual probes, which are in terms of the original probes.

There are limitations to this approach. Beginning with the full set of components can incidentally retain small components and make estimates of beta coefficients unstable[56]. Interpretation of the components is challenging, and here they were generated without the dependent variable (the measurements along the anterior–posterior axes). At the theoretical level PCA can break down when there are many more variables than observations since the sample covariance eigenvectors may not be close to population eigenvectors[57] though empirical results here are positive and in concordance with previous results. Partial least squares (PLS) is a method related to PCR that accounts for the dependent variable and returned similar results (Fig. S4).

As a comparison to a more simple approach, we ranked all probes by the correlation between probe expression and position along the longitudinal axis of the hippocampus. We then selected the top 50 positive and top 50 negative correlations, and selected these features as the top 100 correlation-ranked probes. We assessed overlap between this set and the set of top 100 features identified by our LASSO-PCR model. We also fit a LASSO model and a linear regression model using the following probes as features: (1) top 100 LASSO-PCR features, (2) top 100 PLSR features, (3) top 100 correlation-ranked features. We assessed model accuracy through 10-fold cross validation, and repeated this process with 10 bootstrapped samples to derive confidence intervals. The purpose of this analysis was to see if we gain any predictive accuracy be employing our LASSO-PCR approach vs. a more simple approach for either model fitting or feature selection.

To test the generalizability of the model, we employed several cross-validation methods. First, we performed 10-fold cross-validation of the full data set, which was repeated 10 times. Second, we performed a leave-one-subfield-out cross-validation, to see if a model defined on five hippocampal subfields (CA1-4, subiculum, dentate gyrus) could predict the axis position of samples from the sixth subfield. Finally, we performed leave-one-donor-out cross-validation to see if a model trained on samples from five donors could predict the axis positions of samples from the sixth donor. Note that the range of sample position was constrained by anatomy during the leave-one-subfield-out cross-validation, and the number of samples varied quite dramatically across donors for the leave-one-donor-out validation. The final model used for all subsequent analyses utilized all samples.

### Model feature deconstruction to identify specific genes.
An advantage of the LASSO-PCR model is that it is more likely to identify several genes participating in a co-expression network rather than arbitrarily identifying a single gene to represent that network. However, this also leads to a possible disadvantage related to reduced precision in singling out which genes, if any, are singularly important to the model. Additionally, the global feature importances of a LASSO model cannot be reliably interpreted, as adding or removing features can cause feature importances to shuffle dramatically[21]. We attempted to de-construct our model with these limitations in mind. Fifty probes with, respectively, highest (anterior) and lowest (posterior) back-transformed weight (feature importance) were iteratively removed from our model. After each removal of these 100 probes, the model was refit, 10-fold cross-validation (CV) accuracy was recorded, and the 100 top probes from the new model were removed. This process was repeated until all probes were removed. As a control, we repeated this same process iteratively removing 100 random probes instead of the 100 most important probes. Change in CV accuracy across rounds of probe removal was visually assessed and inflection points were identified at rounds where CV accuracy dropped and did not recover. Rounds in between inflection points were considered stable, and probes removed between inflection points were grouped together in gene sets, and analyzed separately in subsequent analysis.

To establish whether these gene sets alone could predict sample position along the longitudinal axis of the hippocampus, we reran the LASSO-PCR model with only the probes involved in these gene sets. Prediction accuracy was recorded using 10-fold cross-validation. The models were run ten times with bootstrap samples to attain confidence intervals. As a control analysis, models were run using sets of random probes the same size as each gene set, and this process was repeated 10 times for each set, each time using cross-validation to measure prediction accuracy. Finally, in order to compare larger gene sets to Set 1—which contained only 100 probes—we extracted 10 random sets of 100 genes from within each gene set and input these into the model, once again using 10-fold cross-validation to measure prediction accuracy.

To further highlight candidate genes associated with the hippocampal longitudinal axis, we employed the Local Interpretable Model-Agnostic Explanations (LIME) python package (https://github.com/marcotcr/lime/). LIME makes local perturbations to model inputs and measures the impact of those perturbations on model performance. LIME can only assess local feature importance, but by aggregating information across multiple local features, some limited information can be ascertained about contribution of features (probes) to predicting an outcome (sample position along the longitudinal axis). For each gene set identified, we performed 10-fold cross-validation with a Random Forest Regressor. A Random Forest Regressor was chosen because its metric of feature importances is itself assessed using out-of-sample prediction. For each fold, LIME was used to identify absolute feature importances for samples in the left-out fold, and this information was aggregated across all predictions from all folds. Elevated feature importance could indicate importance of a probe across prediction of

multiple samples, or could indicate great importance across a limited set of predictions, meaning interpretation is still limited.

### Gene ontology enrichment analysis.
GO enrichment analysis was used to characterize functions shared by several genes within gene sets. These analyses were performed using the online tool GOrilla (RRID:SCR_006848; http://cbl-gorilla.cs.technion.ac.il/), which identifies terms from the GO libraries that are associated with genes in the inputted gene set and are significantly (FDR < 0.1) enriched compared to a baseline gene set. We used the entire set of genes available in the Allen Human Brain Atlas dataset as the baseline gene set. Altogether, the background set we entered included 29,381 distinct genes, 19,895 of which were recognized by GOrilla. Of these, only 17,836 were associated with a GO term. We left all other parameters to their defaults. Some of the gene sets produced long lists of enriched terms. We summarized this information using hierarchical agglomerative clustering on the significantly enriched terms. A binary gene × term matrix was created where a 1 indicated a gene was associated with a term. This matrix was fed to an Agglomerative clustering algorithm using Jaccard index with average linkage and pre-calculated connectivity constraints (10 neighbors), and the process was repeated varying the number of clusters from 2 to 20. Local peaks in silhouette index were used to define the final cluster number, favoring a higher number of clusters for better precision. The resulting clusters represented sets of genes sharing several associated terms. For gene Set 2 (top 101–600 most important probes to the model, see the section (Genes associated with the long axis of the human hippocampus (in Results)), peaks in Silhouette score were seen at $k = 2$ (0.225), $k = 7$ (0.132) and $k = 10$ (0.128). We chose a 10-cluster solution. For gene Set 3 (top 601–2700 probes, the section (Genes associated with the long axis of the human hippocampus (in Results)), peaks in Silhouette score were seen at $k = 2$ (0.349), $k = 5$ (0.173) and $k = 12$ (0.093). We chose a 12-cluster solution. The purpose of this analysis was to cluster genes with enriched GO terms for purely descriptive purposes.

### Partitioning patterns of anterior–posterior gene expression.
We would expect that the distribution of expression patterns across the longitudinal axis of the hippocampus would be a mix of linear and non-linear patterns, given that both graded and segregated moelcular gradients are at play during the course of development. In order to address this possibility, we assembled all 5000 probes from Gene Sets 1–4 (the features found to be relevant to the longitudinal axis) and performed Spectral Clustering with 10 initial seeds and an radial basis function kernel with a default gamma value of 1. To better prepare the data for clustering, values for each probe were normalized to 0–1 scale, ordered by position from posterior to anterior, and were smoothed with a 3 mm gaussian kernel across this dimension. This process reduces some of the idiosyncrasies of probe-specific variance in expression along the axis. We repeated the clustering analysis varying the number of clusters between 2 and 50. Local peaks in silhouette index were identified at $k = 3$ (0.078), $k = 10$ (0.045), and $k = 14$ (0.041), and a 14-cluster solution was chosen to maximize the number of expression patterns. Mean expression patterns for each cluster were calculated across all features and individually within gene sets, and the cluster membership proportions were calculated for each Gene Set.

### Cell-type analyses.
Many of the genes among our top features are markers for specific cell-types, leading us to question whether the variation in gene expression we observe along the longitudinal axis may be explained by variation in cell type. Cell type characterization is an active area of research, and establishing cell type proportions from bulk microarray data is a complicated task with many limitations. Despite these challenges, we utilize two different protocols to attempt this task, each using a different reference sample.

First, we use the online tool CiberSortX[25] to create a signature matrix from an external sample of single-cell RNAseq data, and use this information to estimate cell fractions from our bulk microarray hippocampal data. Our reference dataset consisted of 15,928 cells extracted from the human middle temporal gyrus, made available by the Allen Brain Atlas https://celltypes.brain-map.org/rnaseq. Each of these cells has been catalogued into one of 169 distinct neuronal and non-neuronal cell types using methods previously described[22]. To ensure the reliability of this approach, we divided the dataset into four subsamples of 4000 cells each and used CiberSortX to create a separate signature matrix for each of these subsamples. For each signature matrix, we used CiberSortX to calculate the estimated cell fractions within the bulk microarray hippocampus samples from our previous analyses. This process lead to, for each subsample, a (hippocampus) sample × cell type matrix indicating the proportion of each sample composed by a given cell type. We then only retained cell types for which the average cross-subsample correlation exceeded 0.7, indicating a consistent relative cell fractionation across runs of CiberSortX. Only 15 cell-types fit this criterion, and only these cell types were used for subsequent analysis. For each cell type, we calculated the correlation between sample position along the longitudinal axis and cell type fraction within that sample. We performed this correlation across each subsample, as well as across the mean of the four cell fraction matrices. These associations were FDR corrected with a Q value of 0.1. In the same manner, we also used ANOVAs to calculate the difference in cell type proportions across hippocampal subfields. For creating signature matrices, CiberSortX was run with quartile normalization disabled and all other parameters with their default settings. For cell fractions, batch correction and quartile normalization were enabled, and all other parameters left to their default settings.

As a way of validating results from the previous analysis, we used a completely different pipeline for determining cell type expression inspired by ref. [26], using a different reference sample. We accessed supplementary data from ref. [24] indicating top genes specifically associated with 35 different cell types. For each cell type, we created a sample x gene expression matrix using the top genes associated with that cell, across all hippocampus samples. We then used this as input to a principal components analysis and stored loadings of each gene on the first component. Finally, we arrived at a cell type expression value for a given sample by finding the weighted mean of the expression of all cell-associated genes, using weights from the PCA. As with the previous analysis, we calculated relationships between cell-type expression and axis position, as well as difference across subfields.

**Model generalization to an external prenatal dataset**. While we used repeat 10-fold cross-validation to validate our model, this approach may be biased to our specific sample of six human brains. Therefore, we sought to generalize the model to an external dataset. We downloaded data from 1203 tissue samples laser microdissected from the brains of four deceased neonatal humans, aged 15–21 post-conception weeks, as part of the Prenatal LMD Microarray portion of the Brainspan dataset[27]. Download links and further information including consent information can be found at http://www.brainspan.org. Metadata for the four donors can be found in Supplementary Data 7. Each tissue sample underwent microarray analysis producing gene expression data for the same 58,692 probes as the Allen Human Brain Atlas. The same procedure as described in the section (Human gene expression data) was used to regress out donor-specific expression patterns from each probe. While no stereotaxic coordinates are available for the current version of the Brainspan atlas, we were able to identify 53 samples falling within the hippocampus using the precise anatomical labels associated with each tissue sample. Specifically, all samples labeled as being within the CA1-4, hippocampus, dentate gyrus, subiculum, presubiculum or postsubiculum were included as hippocampus samples. These labels additionally indicated whether the sample was extracted from the rostral or caudal hippocampus, and so all samples were divided into a rostral or caudal group.

The feature sets (i.e. probes) are identical between the Brainspan and Allen Human Brain Datasets, which allowed us to easily generalize our model to the Brainspan data. For each sample, we calculated the dot product between the vector of probe expression values and the vector of probe weights from the model trained on the hippocampus longitudinal axis, described in the section (Genes associated with the long axis of the human hippocampus (in Methods)). This produced a single value for each sample, representing the pattern expression of the model trained in the Allen Human Brain Atlas dataset. To improve interpretability, the intercept was not added, which effectively normalized the pattern expression. Note that this process is identical to the process to calculate HAGGIS (see below, the section (HAGGIS formulation). Higher values should therefore be mapped onto more anterior samples, while lower values should indicate samples were extracted from a more posterior portion of the hippocampus. To test this, we compared the pattern expression between rostral and caudal hippocampus samples using t-tests. We furthermore used logistic regression to see how well the pattern expression of the model trained in adults could predict samples extracted from the rostral vs. caudal hippocampus of the prenatal dataset, and we used this information to create ROC curves. We also repeated this process for each of the smaller individual gene sets identified in the section (Model feature deconstruction to identify specific genes), by substituting the betas from the model using all features with the betas from the model using only genes in each smaller set.

**HAGGIS formulation**. We sought to ascertain to what degree the specific pattern of genes signatures varying along the hippocampal longitudinal axis was expressed throughout the rest of the brain. The probe weight (beta) vector from the LASSO-PCR analysis can be thought of as a hippocampal longitudinal axis genomic signature. In order to test for the presence of this signature in other brain regions, we found the dot product between the beta vector (genomic axis signature) and the gene expression (probe) vector for each sample (Fig. 5c). Note that when estimating regression coefficients we have:

$$\hat{\beta} = (X^T X)^{-1} X^T Y \qquad (1)$$

This is equivalent to using the estimates of coefficients from the LASSO-PCR model to predict the location of the (non-hippocampal) sample along the hippocampal axis. In practice, this amounts to using the hippocampus model to predict where a non-hippocampus sample might fall along the hippocampal longitudinal axis based on that sample's gene expression. However, conceptually, this value can also offer an index of covariance between a given sample's gene expression and the gene expression profile of the anterior or posterior hippocampus. Higher (positive) values represent greater genomic covariance with the anterior hippocampus, while lower (negative) values represent greater similarity to the posterior hippocampus. For the purposes of parity, this index will be referred to in the text as the HAGGIS.

**Comparisons with resting-state functional connectivity**. For each of the 170 hippocampal samples, a resting-state functional connectivity map was downloaded from Neurosynth (RRID:SCR_006798; http://neurosynth.org/) using the closest available MNI coordinate to the MNI coordinate of the sample. The Euclidian

distance between Neurosynth coordinate and sample coordinate never exceeded 2 mm. Each map is based on the resting-state functional connectivity patterns of 1000 young, healthy individuals from the Brain Genomics Superstruct project[58].

We sought to test whether the genes associated with the longitudinal axis of the hippocampus contribute to the differential brain connectivity observable along this axis. The measurement resolution of resting-state functional magnetic resonance imaging (rsfMRI) limits detail at which differences in connectivity can be observed along a structure as small as the hippocampus. To ameliorate this issue, we divided the hippocampus into genomically determined posterior and anterior subsections, created mean connectivity maps for each, and used these mean connectivity maps to create a subtraction image representing differential functional connectivity between the two poles of the hippocampus[11]. To determine a reasonable division between anterior and posterior hippocampus, we created a split at every point along the hippocampus skeleton. For each split, we classified samples as anterior or posterior based on the position of the coordinate along the longitudinal axis relative to the split. For each split, we next ran Logistic Regression, entering sample class (i.e. anterior or posterior) as the dependent variable and sample HAGGIS as the only independent variable. We then plotted the classification accuracy at each split under the hypothesis that higher anterior–posterior classification accuracy would suggest a more empirically sound anterior–posterior division (Supplementary Fig. 7a). We defined the optimal anterior and posterior cut points as (i) local maxima in accuracy that (ii) were at least 3 mm from both hippocampal poles and (iii) captured at least 20 samples for each side of the split. This lead to an anterior split point of $y = 108$ (MNI: −19) and a posterior point of $y = 94$ (MNI: −35). All samples in between were removed. Results in the main text are reported using this split but, due to the somewhat arbitrary nature of this analysis, results are also reported for several other splits.

Once the anterior and posterior samples had been defined, a mean image was made of the functional connectivity maps corresponding to each anterior and posterior sample, respectively. The posterior map was then subtracted from the anterior map. The resulting image represented relative functional connectivity to the anterior over posterior hippocampus. For each non-brainstem, non-cerebellum sample, a $5 \times 5 \times 5$ mm cube was drawn around the MNI coordinate of the sample. The mean of rsfmri subtraction image values within the cube was calculated, and this value was used as a measure of relative functional connectivity of the sample to the anterior over posterior hippocampus. Finally, we ran a Pearson's correlation between this functional connectivity measure and the HAGGIS. A positive correlation would indicate that brain regions with more genomic similarity to the anterior or posterior hippocampus would be more likely to be functionally connected to those regions, respectively. The residuals of this relationship were stored in order to visualize disagreements between these two cortical patterns. This analysis was performed using weights from the model performed on the entire gene set, as well as weights from models defined on individual gene sets.

We repeated this analysis using three other brain masks: (i) All brain regions; (ii) all regions except cerebellum, brainstem and hippocampus; (iii) cerebral cortex only. In addition, we varied the radius of the cube drawn around the sample coordinate between 1 mm and 6 mm. For completeness, we performed the above analysis using each cube radius, with each mask, and using many different splits—a total of 336 analyses. To ensure the relationships between HAGGIS and rsfMRI connectivity were not born out of chance, we performed a permutation test for each of the 336 conditions. Specifically, the gene expression values for each sample were randomly shuffled, and a correlation was run between the extracted rsfMRI connectivity values and the shuffled gene expression values. This process was repeated 1000 times to create a null distribution, to which the observed value was compared to establish an exact p-value.

We performed one final validation by applying diffusion map embedding[6,29,30] —a non-linear dimension reduction approach—to the hippocampal-brain functional connectivity matrix. This approach summarizes variation in hippocamus-brain connectivity into components or gradients[6], allowing threshold-free representations of variation in hippocampus–brain functional connectivity for each tissue sample. The whole-brain connectivity maps for each sample (see above) were masked with a cortex-only mask (see above), vectorized and concatenated into a Sample × Voxel matrix. A correlation matrix was created from the transpose, generating a Sample × Sample similarity matrix, which was reduced using diffusion map embedding with default settings. We report the total variance in hippocampus–brain functional connectivity explained by each gradient, as well as the $r^2$ summarizing each gradient's relationship to sample location along the longitudinal axis, and predicted sample location based on gene expression (proportionate to HAGGIS). We also report p-values, which are Bonferroni corrected for multiple comparisons. We then selected the gradient with the greatest relationship to predicted sample location (i.e. HAGGIS), provided this relationship was significantly stronger than that of other gradients, as measured using Steiger's tests[6]. For these select gradients, we also report this information with sample location predicted using each of the gene Sets described above (section (Genes associated with the long axis of the human hippocampus (in Results)) and section (Model feature deconstruction to identify specific genes)).

Other studies have been published examining genomic associations with functional connectivity[37,38], and so we sought to understand what proportion of the variance explained from the main analysis (shown in Fig. 5a) was unique to the HAGGIS rather than general network connectivity. We trained a cross-validated PLS model to learn the genomic features predicting relative anterior vs. posterior connectivity to the

**Table 2 Demographic information for FTD and AD individuals included in FDG-PET analysis.**

|  | AD | FTD | Test |
|---|---|---|---|
| Age at FDG: mean (sd) | 62.0 (8.8) | 61.4 (8.7) | Cohen's $d = 0.07$, $p(t\text{-test}) = 0.79$, $p(\text{Mann–Whitney}) = 0.82$ |
| Females: $n$ (%) | 12 (34%) | 19 (54%) | Fisher exact $p = 0.15$ |
| Years of education:mean (sd) | 16.1 (2.9) | 16.3 (4.7) | Cohen's $d = 0.04$, $p(t\text{-test}) = 0.88$, $p(\text{Mann–Whitney}) = 0.81$ |
| Dementia stage (CDR ≥ 1): $n$ (%) | 22 (63%) | 17 (49%) | Fisher exact $p = 0.34$ |
| CDR-SoB: mean (sd) | 4.8 (1.9) | 4.2 (3.2) | Cohen's $d = 0.25$, $p(t\text{-test}) = 0.30$, $p(\text{Mann–Whitney}) = 0.31$ |

FDG fluorodeoxyglucose, sd standard deviation, CDR clinical dementia rating, CDR-SoB clinical dementia rating, sum of boxes.

hippocampus (i.e. the map in Fig. 5a; see the subsection (Identifying candidate genes for brain-hippocampus interactions) below for details). We considered the 10-fold cross-validated variance explained of this model to represent an estimate of the maximum variance explainable given the present genomic data. We then represented the variance explained of HAGGIS as a proportion of the overall variance explainable given the genomic data (visualized in Fig. 5c).

**Comparisons with structural covariance**. Structural covariance is thought to reflect shared cytoarchitecture and/or developmental and degenerative trajectories between regions[59]. The anterior and posterior hippocampus have shown different patterns of structural covariance with the rest of the brain[28], and structural covariance appears to be genetically determined to some extent[59]. Accordingly, we assessed whether the differential structural covariance between different brain regions and the hippocampus along its longitudinal axis is reflected by patterns of genomic covariance.

Structural covariance was calculated using the OASIS: Cross-Sectional structural (T1) MRI dataset[60], accessed with Nilearn (RRID:SCR_001362; https://nilearn.github.io/). The OASIS images came preprocessed using the SPM DARTEL pipeline[61]. 153 preprocessed gray matter volume images were identified as healthy, cognitively normal young (age < 40) controls. For each voxel corresponding to the MNI coordinates of an Allen Human Brain Atlas hippocampus sample, a structural covariance vector was calculated between that voxel and all other brain voxels. Elements in the vector represented Pearson correlation coefficients between voxel values across the dataset of 153 individuals between the two regions. Anterior and posterior hippocampus divisions identified in the previous analysis were used to divide the covariance vectors, and the average covariance within anterior vectors and posterior vectors were calculated, respectively. The difference between these vectors was calculated to create a map where each voxel contained a value representing the relative structural covariance to the anterior over the posterior hippocampus. The values strongly favored the anterior hippocampus, so the map was z-scored, such that lower values represented less structural covariance to the anterior hippocampus. Relationships between HAGGIS and relative structural covariance were carried out in a manner identical to the functional connectivity analysis described above, and were repeated using different gene sets and brain masks. The residuals of this relationship were stored in order to visualize disagreements between these two cortical patterns. Similar to the functional connectivity analysis, we calculated the variance explained by HAGGIS as a proportion of the maximum variance explainable given the data (see the section (Comparisons with resting-state functional connectivity)).

As with the functional connectivity analysis, we used diffusion map embedding to generate threshold-free measures (gradients) summarizing structural covariance between the hippocampus and other parts of the brain. For each sample, we calculated structural covariance between the voxel at the sample location and all other voxels falling within a cortical mask, creating covariance vectors. These vectors were concatenated into a Sample × Voxel matrix, and reduced using diffusion map embedding as described above (section (Comparisons with resting-state functional connectivity)).

**Comparisons with neurodegenerative disease vulnerability**. Previous studies have noted the differential relationship of the hippocampus to AD and FTD. We tested whether regions more genomically similar to the anterior than posterior hippocampus might be more vulnerable to neurodegeneration in FTD than in AD (and vice versa). In April 2018, we queried our database looking for patients who fulfilled the following criteria: (i) Had available both a [$^{11}$C] Pittsburgh Compound B (PiB)-PET scan for β-amyloid and a [$^{18}$F] Fluorodeoxyglucose (FDG)-PET scan of brain glucose metabolism acquired on the Biograph scanner; (ii) Had either a clinical diagnosis of AD[62] and a "positive" PIB-PET read, or a clinical diagnosis of FTD (either behavioral variant FTD or semantic variant primary progressive aphasia, as described in ref. [63]) and a "negative" PIB-PET read. This query resulted in 36 AD and 39 FTD patients. Five patients were later excluded because of incomplete FDG-PET SUVR (missing at least one of the six frames between 30 and 60 min post injection), resulting in a final count of 35 AD and 35 FTD patients. Demographic information can be found in Table 2. Note there is no overlap between this sample and the sample described in ref. [11].

All patients were seen at the at University of California, San Francisco Memory & Aging Center and imaged at the Lawrence Berkeley National Labs. Informed consent was obtained from all subjects or their assigned surrogate decision-makers, and UCSF and the Lawrence Berkeley National Laboratory (LBNL) institutional review boards for human research approved the study. PET acquisition details can be found elsewhere[64]. FDG-PET images were processed using SPM12 using a previously described pipeline[64]. Briefly, six five-minute frames were realigned and averaged, and the average image was coregistered onto patient specific anatomical T1-MRI scans. Standard uptake value ratios (SUVR) were calculated using the pons (Freesurfer segmentation of the brainstem with manual cleaning) as a reference region, and SUVR images were warped to the MNI template using MRI-derived parameters. All 70 patients were entered into a voxelwise t-test controlling for age and disease severity (Clinical Dementia Rating Sum of Boxes score) using SPM12, highlighting differences in glucose hypometabolism (a proxy for neurodegeneration) between AD and FTD patients. The t-map from this analysis was used for subsequent analyses, and is made available with this publication (https://neurovault.org/collections/4756/).

For each non-brainstem, non-cerebellar sample, a 5 mm diameter cube was drawn around the sample's MNI coordinates, and the mean t-value from the t-map described above was extracted. This value represents the relative neurodegeneration in FTD over AD in or around the region the sample was extracted from. Across samples, a correlation was calculated between this value and the sample's HAGGIS. A positive correlation would suggest regions more genomically similar to the anterior than the posterior hippocampus are more vulnerable to neurodegeneration in FTD than in AD. The residuals of this relationship were stored in order to visualize disagreements between these two cortical patterns. To ensure our findings were not specific to the brainmask used or the size of the extraction cube, we reran the analysis using each of the three additional masks described in the section (Comparisons with resting-state functional connectivity), as well as varying the diameter of the extraction cube. Finally, permutation tests were run for each condition to compare our observations to chance (see the section (Comparisons with resting-state functional connectivity)).

As with the previous analyses, we ran the above analyses across different gene sets. Finding divergence across sets, we performed post hoc analyses to determine whether gene ontology analysis could explain this discrepancy. Using gene ontology clusters defined in the section (Gene ontology enrichment analysis), we created cluster scores for Gene Sets 2 and 3. Specifically, we calculated the weighted mean expression of genes associated with each gene ontology cluster where weights were determined by cluster centrality. We next found the correlation between each cluster score and relative disease vulnerability across all brain samples. For each cluster score, we also shuffled the weights of the weighted mean and reran the correlations with disease vulnerability 100 times, creating a null distribution. Associations greater than the 95% of null distribution values were considered to be greater than chance.

**Identifying candidate genes for brain-hippocampus interactions**. In the sections (Comparisons with resting-state functional connectivity), (Comparisons with structural covariance), and (Comparisons with neurodegenerative disease vulnerability), we describe methods to uncover relationships between HAGGIS and hippocampus–brain interactions. We wished to identify which specific genes were principally involved both in the organization of the longitudinal axis of the hippocampus, as well as in the hippocampus–brain interactions, further elucidating the role of the various genes identified in the section (Model feature deconstruction to identify specific genes) along the axis. For each hippocampus–brain interaction map (visualized in Fig. 5a), we fit a partial least squares (PLS) regression model with gene expression information as X and hippocampus–brain interaction value as Y, across all brain samples. As with the model described in the section (Genes associated with the long axis of the human hippocampus (in Methods)), the X input was first transformed using principal components analysis and represented as a set of genomic components. The model was fit varying the number of PLS components (i.e. modes) between 1 and 10, and using 10-fold cross-validation to assess model accuracy. The model with the highest cross-validated explained variance was selected as the best model, and was considered the maximum explainable variance given the genomic data available, which was therefore useful to compare to the HAGGIS models (see the section (Comparisons with resting-state functional connectivity) above). Note that the hippocampus itself was not included in any of the models. For each of the three PLS models, feature weights were backtransformed back into probe space (see the section (Genes associated with the long axis of the human hippocampus (in Methods)), and the top 50 anterior and posterior

associated features (i.e. with the highest and lowest weights) were identified. Overlapping features between each model and the hippocampus longitudinal axis model are reported. These features represent genes that appear to be very important in predicting the location of tissue samples in the hippocampus, but also in predicting interactions between the hippocampus and other brain regions. To ensure this overlap did not occur by chance, 1000 sets of 100 random probes were generated, and used to calculate the probability of overlap between 100 random features and the 100 features from the the hippocampus longitudinal axis model.

**Comparisons with large-scale cognitive systems.** Neurosynth contains 3D meta-analytic functional co-activation maps from task-fMRI studies that are paired with sets of related topics (words) extracted from the text of these studies. These topic-list/co-activation map pairs are the result of a Latent-Dirichlet Allocation across 11,406 articles, the details of which can be found elsewhere[65]. In short, topic lists represent words that are mentioned greater than chance (FDR < 0.01) in papers reporting functional co-activation in given coordinates, summarized by paired co-activation maps. All 100 (association/reverse inference) maps from the set of 100 topic-list/co-activation map pairs on the Neurosynth website were downloaded and binarized such that all values above 0 were set to 1, and all other values were set to 0. We manually labeled the topics according to their hypothesized association with the AT/PM system[10] based on the content of the word list (AT/PM/Not associated) but without reference to the spatial pattern of the co-activation. For each of the 100 binarized functional meta-analytic co-activation maps, all samples with MNI coordinates falling within the map were identified, and the mean HAGGIS of those samples was calculated. Therefore, each topic/map pair had an associated value indicating the degree to which the brain regions involved expressed genes similar to the anterior or posterior hippocampus. Higher values represented similarity to the anterior hippocampus, lower values to the posterior hippocampus, and higher absolute values represented greater genomic covariance. To increase confidence in this approach, the main analyses were restricted only to maps overlapping with at least 500 samples (29/100).

To help visualize these results, we created a word cloud summarizing both the spatial (functional coactivation) and topic (cognitive) information associated with the anterior and posterior hippocampus respectively. For the topic information, each topic-set contained 40 words arranged by importance to the topic-set. Each word was given a value proportionate to its importance rank in its topic set (i.e. most important word valued at 40, least important at 1). Next, the value of each word was multiplied by the average HAGGIS within the binarized map paired to the word's topic-set (i.e. the bars in Fig. 6), multiplied by 1000 to increase the weighting of this multiplier proportionate to the within-set ranking. Therefore, each word had an associated value, such that the highest values represented words most important to topic/map pairs with the greatest HAGGIS, where multiple mentions increased the value of the word. To summarize the spatial information, we binarized each map and multiplied it by the average HAGGIS within the binarized map (i.e. the bars in Fig. 6), and summed all maps, and smoothed the image with a 4 mm isotropic kernel. All voxels with positive values were binarized into a mask, and this mask was used as constraint for the anterior–hippocampus word cloud, inside which the top 100 words were visualized. All voxels with negative values were binarized into a posterior mask used as a constraint for the posterior–hippocampus word cloud. The word values were repeated inverting the HAGGIS multipliers, and the top 100 words were visualized. The final image represents brain regions coactivated more with the anterior vs. posterior hippocampus, and the cognitive topics most associated with those regions.

**Reporting summary**. Further information on research design is available in the Nature Research Reporting Summary linked to this article.

## Data availability
All data used in this paper are publicly available. A summary of datasets used including access links can be found in Supplementary Data 8. Information about ethical compliance and consent for each dataset can be found by following the respective links. The FDG difference map between AD and FTD patients has been deposited in Neurovault (https://neurovault.org/collections/4756/). All other figures can be generated de novo using scripts and data provided at https://github.com/illdopejake/Hippocampus_AP_Axis[51].

## Code availability
All data and analyses described in this paper are available online and can be fully reproduced using exclusively open-access software, with python scripts and data provided at https://github.com/illdopejake/Hippocampus_AP_Axis[51]. All code and analyses are presented in a series of Jupyter notebooks at the link provided. Supplementary Data 9 outlines which notebook contains the analyses described in each subsection of the Methods.

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

## Acknowledgements

We would like to thank David Berron, Konrad Wagstyl, Mallar Chakravarty, J. B. Poline, Jakob Seidlitz, Kevin Anderson, and Gabriel Devenyi for advice and recommendations. We would also like to acknowledge support from the Ludmer Centre for Neuroinformatics and Mental Health and the Healthy Brains for Healthy Lives initiative. Additionally, we thank Bruce Miller, Howie Rosen, and Bill Jagust for supporting the FDG studies in AD and FTD, which were funded through the following sources: NIA R01 AG045611 (Rabinovici), P50 AG23501 (Miller, Rosen, Rabinovici), P01 AG019724 (Miller, Rosen, Rabinovici). J.W.V. is funded through a Tri-Council Vanier CGS Doctoral Fellowship. R.L.J. is funded through an Alzheimer's Association Research Fellowship (AARF-16-443577, PI: Renaud La Joie). B.B. acknowledges research funding from CIHR, NSERC, SickKids Foundation, the Azrieli Center for Autism Research, and salary support from FRQS. R.V.D.W. was funded through a Savoy Foundation fellowship. Nearly all data and tools used in this paper are publicly accessible, and this work would not be possible without the hard work put into collecting, storing and maintaining these data and platforms. Data were provided in part by OASIS: Cross-Sectional: Principal Investigators: D. Marcus, R, Buckner, J, Csernansky J. Morris; P50 AG05681, P01 AG03991, P01 AG026276, R01 AG021910, P20 MH071616, U24 RR021382. Data were additionally provided in part by the Brain Genomics Superstruct Project of Harvard University and the Massachusetts General Hospital, (Principal Investigators: Randy Buckner, Joshua Roffman, and Jordan Smoller), with support from the Center for Brain Science Neuroinformatics Research Group, the Athinoula A. Martinos Center for Biomedical Imaging, and the Center for Human Genetic Research. Twenty individual investigators at Harvard and MGH generously contributed data to GSP Open Access Data Use Terms Version: 2014-Apr-22 the overall project. Data were additionally provided from numerous studies included in Neurosynth.

## Author contributions

J.W.V., M.J.G., and R.L.J. conceptualized the study. J.W.V., A.D.P, A.D., E.V.P., C.L., R.V. D.W., and B.B. designed and developed the methodologies. J.W.V. analyzed the data. R.L. J. and G.D.R. provided patient data. J.W.V., R.L.J., M.J.G., and A.D.P wrote the paper. J.W.V., R.L.J., R.A.T., Y.I.M., and B.B. interpreted the findings. All authors revised the paper and provided critical feedback. R.L.J., M.J.G., B.B., and A.C.E. supervised the study.

## Competing interests

The authors declare no competing interests.
