## [Peer Review File · Nature Communications]

Reviewers' comments:

Reviewer #1 (Remarks to the Author):

Vogel et al. present a study of genomic anatomy along the longitudinal axis of the neurotypical adult human hippocampus. The authors leverage publicly available data to develop models that can accurately predict longitudinal hippocampal position based on gene expression. After identifying the genes that contribute most informatively to these models, the authors explore how differences between anterior (A) and posterior (P) hippocampal gene expression relate to gene expression in other human brain regions (termed 'HAGGIS'). They then proceed to identify brain regions with differential resting state functional connectivity, differential structural covariance, or differential vulnerability to AD or FTD (as measured by FDG-PET) with respect to transcriptionally defined A or P hippocampus. Lastly, the authors examine how variation in A-P genomic signatures (HAGGIS) relate to distributed cognitive networks.

I enjoyed reading this study, which I thought was creative, ambitious, and well written. The authors' motivations were clear and the analyses were rigorous and clearly described. I have only a few comments and questions for the authors, which are listed below:

- 1) I would have liked to have seen more discussion / data regarding identified genes (Table 1) and their relation to variation in cellular composition along the longitudinal axis. For example, one of the top anterior genes is RSPH9. In the Discussion, the authors mention that this gene is "part of the structure of primary cilia, which can be found within ependymal cells lining the ventricles" (this sentence is a bit confusing, since most RSPH9 expression in ependymal cells is likely to support motile cilia, which are far more abundant). This finding begs the question of whether other anterior genes may also mark ependymal cells, suggesting a greater contribution of ventricular surface in samples from A vs. P hippocampus. Similarly, PVALB marks a class of interneurons that may be differentially abundant in A vs. P hippocampus, and TTR marks choroid plexus cells. It would be interesting to cross-reference genes from Table 1 with cell type-specific genes from single-cell or other resources to see if any patterns emerge.
- 2) There are a number of places in the manuscript where the authors suggest that genes in Sets 1, 2, or 3 are 'regulating' (p. 9), 'playing a significant role in the positioning' (p. 29), or 'responsible' (p. 31) for longitudinal axis positioning of the hippocampus. This kind of causal language struck me as a bit awkward since the patterning of the axis is established during early brain development and the authors have not demonstrated that any of these genes are required to maintain A-P patterning.
- 3) I found the data in Fig. 1H to be slightly confusing (particularly, the "all" sets). Does it make sense to color these data points orange and blue in the same way as Sets 1-4?
- 4) The study would be strengthened with histological validation of candidate genes from Table 1 in independent human hippocampal samples, but I don't think this is critical.
- 5) On p. 27, I suggest changing "This analysis provides an estimate of the relative expression of different proteins..." to 'transcripts'.
- 6) I think the authors have used a defensible approach to identify their top 100 genes in Table 1, but did they compare their results with much simpler strategies (e.g. identifying the top 100 genes ranked by their |correlation| with longitudinal axis position)?

Reviewer #2 (Remarks to the Author):

Vogel and colleagues present an interesting set of findings pertaining to the molecular organisation of the human hippocampal long-axis. An excellent, concise introduction is followed by several methodological approaches to characterising long-axis organisation based on data from six deceased human donors from the Allen Human Brain Atlas. The location of hippocampal tissue samples can be correctly ascribed to their position of origin along the long-axis with accuracy. Notably, the expression pattern of only a fraction of the genes studied (100 genes of 58,692) was required to reach high accuracy. Of these 100, Random Forest Regression picked out just a handful that was important for prediction, with some genes showing differential predictive value for different subfields. The results of the first analysis, a regularized PCA approach, are shown to converge with a second approach (another regression method based on dimension reduction; partial least squares regression). The authors then looked further afield from the hippocampus and derive an index of similarity between the genomic signature of the hippocampal longitudinal axis and that of samples from the rest of the brain (called HAGGIS). Sampling is obviously more sparse, but there is some overall trend for differences between anterior vs posterior hippocampus-associated gene expression. The midline effects are most striking, but it is perhaps difficult to understand the significance of the similarity between anterior hippocampal genetic signature and that of the extent of the dorsal aspect of the brainstem. HAGGIS from each sample was then regressed against other measures – resting-state functional connectivity, structural covariance, and glucose PET comparing AD vs frontotemporal dementia derived from open access databases – each showing a positive effect. The limitations section covers the obvious short-comings of the work, such as sample size.

These results are an important addition to a growing literature on the genetic architecture potentially underlying hippocampal long-axis organization that has until now been studied in the rodent. I have some comments for the authors to address, in order of importance.

- With respect to interpretation of results in terms of the AT-PM dissociation, I am not sure how well the data fits this theory. AT-PM was proposed to account primarily for perirhinal/parahippocampal dissociations. The AT system comprises lateral orbitofrontal and ventral temporopolar cortex and amygdala; while the PM system includes posterior cingulate, precuneus, angular gyrus, ventromedial prefrontal cortex. Thus, there are key anatomical differences that should not be overlooked, both in terms of inconsistencies and omissions. For the former, the vmPFC and angular gyrus are linked via HAGGIS to anterior hippocampus (predicted to be posterior and part of PM system). For the latter, how can the prominent brainstem and cerebellar relationships with anterior and posterior hippocampus, respectively, be reconciled with this account?

A linear division between brain areas – following the axis of the brain stem – seems like the most parsimonious segregation.

- A conceptual point to make is that there appear to be a combination of effects here, with some genes expressed almost selectively in posterior vs anterior regions, while others show more of a graded pattern. This is clear from Fig 2B (note that the x-axis should be labelled in this figure). The authors are aware of this, and dedicate Supp Fig 4 to show graded organisation. Given that gradients are a core feature of the manuscript, I would consider including this figure in the main text. The co-existing patterns of segregated vs graded gene expression should be stressed, as it concurs with rodent findings.

- In the abstract the authors state that “posterior parieto-occipital and cerebellar regions involved in spatial cognition, selectively vulnerable to Alzheimers disease,..” I would not say that these regions are selectively vulnerable to AD. Taking the cerebellum as an example. Regardless of whether one thinks that the cerebellum is an “innocent bystander” in AD or not, or whether it is less affected by AD pathology than other regions, the cerebellum is also affected in FTD. Indeed, the paragraph in the discussion (from line 467) dedicated to AD and FTD is, in my opinion, too speculative. After all, if AD pathology is considered to begin within locus coeruleus and transentorhinal cortex, these are areas showing genetic similarity with anterior hippocampus. I appreciate that this comes down to a question of the primary pathogenic process in question, but tau pathology in AD has not been considered in the authors' discussion.

- I may have missed this somewhere, but demographics for the post-mortem individuals (e.g. age, gender, cause of death, post-mortem interval) should be provided.

Minor points

- In the results section, when referring to genes previously described in studies exploring the rodent longitudinal axis, please reference the relevant studies. Also, please report first the 6 subfields used, before describing results associated with them.
- P. 28 methods: "all samples across the six donors were aggregated, the effect of donor was removed from each probe using linear models (i.e. with dummy coded donor ID variables)". Does this really "remove" the effect of age/gender etc., or just model the linear interactions between the tested variable and these parameters?
- Legend for Figure 1 "G) The first 50 rounds of 100-probe removal from Panel A. Inflection points were identified after removing 100, 600, and 2700 genes." Should read Panel F
- Fig 2, Supplementary Fig. S2 "Contribution" and "Importance" are used interchangeably and might confuse some readers

Reviewer #3 (Remarks to the Author):

In their manuscript titled "A molecular gradient along the longitudinal axis of the human hippocampus informs large-scale behavioral systems," the authors find that the differential connectivity and behavioral roles of anterior and posterior hippocampus in human is reflected in a transcriptomic signature along the longitudinal axis. This work builds on similar findings in rodent by leveraging existing human genomic and imaging data sets and shows that this hippocampal axis is a conserved feature of mammalian brain across >65 million years of evolution. This work provides a useful statistical framework (and code) for identifying specific genes contributing to expression gradients in general and will be useful to the community. A major limitation of the work is that it relies on microarray data from bulk samples that include a diversity of neuronal and non-neuronal cell types. However, the authors help mitigate this by considering dissections from different hippocampal sub-fields separately, and the results will provide a useful comparison for future single cell RNA-seq analysis from this brain region.

Specific comments:

- Please report which genes that have been reported to have a dorsal/ventral gradient in mouse appear (or not) in Table 1. Are there any genes that change polarity, e.g. are dorsally and anteriorly enriched?
- Can you comment on whether you think these gradients are driven by changes in cell type proportions or expression within cell types? Which cell classes might be involved? You can compare to single nucleus RNA-seq from mouse hippocampus or human cortex to guide this analysis.
- You identified thousands of genes that appear to contribute to longitudinal position but only highlighted a small number of genes from the first set and that emerged from the Random Forest analysis. Do you think that only a small number of genes are relevant for setting up the appropriate connectivity and lead to cell type specific susceptibility to neurodegenerative disease? Please add more on this topic to the discussion.
- Fig. 2B – Most of these genes appear to have relatively smoothly increasing anterior/posterior expression. If you correlate gene expression with anterior/posterior position, do genes with strong correlations all appear in your LASSO model? Are there genes in sets 2-4 that show low correlations with A/P position? For example, are there genes with high expression in the body and low expression in head and tail?
- Fig. 4A – Sensorimotor cortex appears to have different functional connectivity and disease involvement than predicted by HAGGIS. Can you comment on some of these regional discrepancies and how one can interpret them?
- Fig. 4B – It is interesting that Set3 is more predictive than Set1 for disease. Is there any GO enrichment for these genes that help explain this result?

Trygve Bakken

Dear Reviewers,

We appreciate the opportunity to revise our manuscript, and we are grateful for the time and effort spent on crafting very thoughtful and constructive reviews. Guided by these comments, we have added several new analyses and figures and have rewritten the manuscript into what we believe to be an improved product.

Recurring topics across the Reviewers included an interest in cell-type variation across the longitudinal axis of the hippocampus, a deeper dive into the expression patterns of top genes along the long axis, and more detailed discussion of disagreements between our described “HAGGIS” index of shared hippocampal-cortical gene expression profiles and other hippocampo-cortical systems. We have addressed each of these points, as well as many others, in detail below. In summary, we have added a cell-type analysis and, other than a subtle variation of astrocytes, we do not find strong evidence for variation in the examined cell types along the longitudinal axis. We also show that many of our top genes show a highly linear gradient of expression across the axis, but other patterns are present as well, and selecting only genes with highest linear correlation with axis position leads to worse models. We have also added several points into the Discussion regarding a) agreements and disagreements between HAGGIS and the AT-PM system, b) the relation of HAGGIS with neurodegenerative diseases, and c) a potential explanation of the strong divergence in HAGGIS observed between the brainstem and cerebellum. Finally, we have reproduced our findings in a completely independent dataset of samples extracted from the hippocampus of prenatal humans, showing not only that our model generalizes well to unseen data, but also suggesting a role for our top features during early stages of brain development.

Altogether, we hope the Reviewers agree that these and other additions have led to a better and more interesting manuscript. We once again thank all in participation for a fascinating and constructive review.

--The authors

Reviewers' comments:

Reviewer #1 (Remarks to the Author):

Vogel et al. present a study of genomic anatomy along the longitudinal axis of the neurotypical adult human hippocampus. The authors leverage publicly available data to develop models that can accurately predict longitudinal hippocampal position based on gene expression. After identifying the genes that contribute most informatively to these models, the authors explore how differences between anterior (A) and posterior (P)

hippocampal gene expression relate to gene expression in other human brain regions (termed 'HAGGIS'). They then proceed to identify brain regions with differential resting state functional connectivity, differential structural covariance, or differential vulnerability to AD or FTD (as measured by FDG-PET) with respect to transcriptionally defined A or P hippocampus. Lastly, the authors examine how variation in A-P genomic signatures (HAGGIS) relate to distributed cognitive networks.

I enjoyed reading this study, which I thought was creative, ambitious, and well written. The authors' motivations were clear and the analyses were rigorous and clearly described. I have only a few comments and questions for the authors, which are listed below:

1) I would have liked to have seen more discussion / data regarding identified genes (Table 1) and their relation to variation in cellular composition along the longitudinal axis. For example, one of the top anterior genes is RSPH9. In the Discussion, the authors mention that this gene is “part of the structure of primary cilia, which can be found within ependymal cells lining the ventricles” (this sentence is a bit confusing, since most RSPH9 expression in ependymal cells is likely to support motile cilia, which are far more abundant). This finding begs the question of whether other anterior genes may also mark ependymal cells, suggesting a greater contribution of ventricular surface in samples from A vs. P hippocampus. Similarly, PVALB marks a class of interneurons that may be differentially abundant in A vs. P hippocampus, and TTR marks choroid plexus cells. It would be interesting to cross-reference genes from Table 1 with cell type-specific genes from single-cell or other resources to see if any patterns emerge.

Response: We thank the Reviewer for this informative and thoughtful comment, which converges with Reviewer 3 Comment #2. The Reviewer's point about some of the top genes from our model marking specific cell types lends to the compelling question of whether our gene expression signature is actually pointing to variation in cell type along the longitudinal axis of the hippocampus. Single cell identification and characterization is a subject of ongoing research, and identifying cell fractions based on bulk microarray data represents a significant challenge for which no standardized and multiply validated protocol has emerged. Despite these challenges, we attempt to characterize cell-type variation along the axis of the hippocampus by combining results across two different pipelines and reference datasets.

We first utilize CiberSortX, an online tool developed to create signature matrices from RNAseq data, and use this information to determine cell fractions from a bulk sample of RNA or microarray data (Newman et al. 2019, Nature Biotechnology). For a reference matrix, we use single cell RNAseq data extracted from the human middle temporal gyrus (MTG) – data downloaded from

the Allen Brain Atlas. 15,928 MTG cells were available, and these cells have been catalogued into 169 distinct cell types using methods that have been previously described (Hodge et al., 2019 BioRxiv). In order to increase the reliability of our analyses, we split the data into four separate sets of ~4000 cells each. For each set, we used CiberSortX to create a signature matrix from the RNAseq data and subsequently establish an estimation of the fraction of each cell type composing each of our hippocampus samples. We then removed cell types that were inconsistent (mean $r < 0.7$) between the four CiberSortX runs, since these should theoretically be similar across subsamples. Only 15 cells fit this criterion, and their average fractionation is visualized below.

After averaging the cell fractions across the four runs, we found three cell types to vary significantly along the longitudinal axis of the hippocampus after a liberal correction for multiple comparisons ($FDR < 0.1$): One astrocyte, one microglia and one excitatory neuronal cell type (see Figure below). All types became more prevalent in more anterior parts of the hippocampus. In contrast, cell type varied highly across hippocampal subfield (Supplementary Table S5).

To try to validate these findings, we used a different approach to establish cell types, inspired by Seidlitz et al., 2019 BioRxiv, using a different reference dataset of 35 cell types taken from Lake et al., 2018 Nature Biotechnology. Specifically, for each sample, we calculated a weighted mean of the expression of top genes associated with each cell type, where the weights were established through cell-type specific PCA across all samples. Using this more coarse approach, six cell types were significantly associated with hippocampal axis position after liberal corrections ($FDR < 0.1$): five excitatory neuronal types and one astrocyte subtype. All cell types but one (an excitatory neuronal type) were more prevalent toward the anterior hippocampus. However, based on comparisons of Lake et al. and Hodge et al. (Fig S4 in Hodge et al.), the excitatory cell from the CiberSortX analysis was not one of those found in the present analysis. Therefore, the only cell type to replicate across the two analyses was the astrocytic cell, and these relationships are visualized in panel B of what is now supplementary Figure S5, shown below:

Based on these analyses, we have found evidence for a weak posterior-to-anterior increase of astrocyte prevalence along the longitudinal axis of the hippocampus, replicated across two different methods with different reference samples. This finding does corroborate previous histological studies of the murine dentate gyrus (Ogata & Kosaka, 2002, Neuroscience). Despite the presence of several genes marking distinct cell types and cell classes among the top features of our axis position model, our analyses suggest that the gene expression differences along the longitudinal axis are not primarily driven by distinct cell types, and that gene expression within cell-type may be varying instead. However, due to the preliminary nature of these analyses and the limitations presented by our data and methods, our conclusions remain tentative and would require more thorough and dedicated study to confirm.

We have summarized our methods, findings and interpretations in new sections in the Methods (Section 8.6), Results (Section 2.3) and Discussion (In 535-545) sections, respectively. Additional information can be found in Supplementary Table 5.

2) There are a number of places in the manuscript where the authors

suggest that genes in Sets 1, 2, or 3 are ‘regulating’ (p. 9), ‘playing a significant role in the positioning’ (p. 29), or ‘responsible’ (p. 31) for longitudinal axis positioning of the hippocampus. This kind of causal language struck me as a bit awkward since the patterning of the axis is established during early brain development and the authors have not demonstrated that any of these genes are required to maintain A-P patterning.

Response: We thank the Reviewer for raising this important point. While we were able to replicate our findings in a tissue samples from prenatal human hippocampi (see response to Point 4 below), we agree our analyses are associative in nature and thus do not support causal language. We have changed the text appropriately for the specific phrases the Reviewer has mentioned, as well as elsewhere throughout the manuscript, resulting in a total of 18 text changes.

3) I found the data in Fig. 1H to be slightly confusing (particularly, the “all” sets). Does it make sense to color these data points orange and blue in the same way as Sets 1-4?

Response: We apologize to the Reviewer that the information represented in this subpanel is unclear. We feel coloring the datapoints in similar colors across all conditions is warranted, as the only thing that changed across conditions is the gene set entered into the model. A quick explanation: accuracy decreases across conditions because less and less relevant genes are contained across conditions (blue). This is particularly notable when we only select random subsamples of 100 features within each set (green), further indicating the importance of the 100 features in Set 1. However, accuracy increases across conditions when selecting random subsets of genes (orange). The random subsets are the same size as the Gene Sets, so a large random sample is more likely to contain more genomic features that are actually related to sample position along the axis. To clarify the point that the random sets are the same size as its paired gene set, we have relabeled this color to “Random Sets (same n).”

We do agree with the Reviewer that the orange datapoint for the “all” sets is redundant – taking random samples of the whole set is akin to simply shuffling the data and running it again, no change should be expected (nor is any change observed). We had originally maintained this for consistency, but to avoid confusion, we have now removed the orange datapoint for “all sets”, and we have also relabeled this condition to “original model”, for clarity. The green datapoint for the “all” set is exactly the same as in other sets -- a random sampling of 100 genes from within the set, and therefore we have elected to keep this datapoint as is.

The new figure can be found below:

4) The study would be strengthened with histological validation of candidate genes from Table 1 in independent human hippocampal samples, but I don't think this is critical.

Response: We agree with the Reviewer that this would be a convincing validation of our findings, though our labs are unfortunately not equipped for histological work. However, we believe the spirit of the Reviewer's comment is rooted in the overall desire for our results to be validated on an external sample. Therefore, we obtained data from the Brainspan dataset, which included identical microarray data to the Allen Human Brain Atlas for 53 samples extracted from the hippocampus of four deceased prenatal humans, aged 15-21 post-conception weeks (see metadata in Table S7). These samples did not have stereotaxic coordinates, but had labels indicating whether they were taken from the rostral or caudal portion of the hippocampus. We used our model trained on samples from the adult hippocampus to predict location of samples in the neonatal hippocampus.

Across all gene sets used, rostral hippocampus samples were predicted to be significantly more anterior than caudal hippocampus samples. In addition, the model trained in adults could discriminate rostral hippocampus samples from the caudal hippocampus samples in the neonate with an accuracy ranging from 66% to 83% (AUC 0.70 – 0.95). We were therefore able to validate our model by generalizing it to a totally separate human sample. Furthermore, the fact that these humans were still undergoing neural development further suggests the genes we identified may be associated with the long axis even during early stages of development.

The manuscript has been updated to reflect these new analyses, with changes made to the Methods (Section 8.7), Results (2.4) and Discussion (ln 422-423). In addition, a new Figure has been added to the main text (Fig. 3), reproduced below.

5) On p. 27, I suggest changing “This analysis provides an estimate of the relative expression of different proteins...” to ‘transcripts’.

We agree with the Reviewer that this is a more appropriate terminology. We have enacted the suggested change.

6) I think the authors have used a defensible approach to identify their top 100 genes in Table 1, but did they compare their results with much simpler strategies (e.g. identifying the top 100 genes ranked by their |correlation| with longitudinal axis position)?

We thank the Reviewer for this interesting suggestion. As the Reviewer suggested, we identified the top 100 probes as ranked by their correlation with longitudinal axis position. 26 probes appeared in both this list and our original list of top 100 genes, suggesting many of the genes identified by our more complex model are simply linearly correlated with axis position. This list has been added to Supplementary Table 1, and the overlap with the top genes from our original LASSO-based model, and those from the PLS model, is recorded in

Supplementary Figure 3D. A few details have also been added to the Methods and Results.

There is also perhaps a practical aspect to the Reviewer's suggestion, regarding whether any advantage was gained by using a machine learning model rather than a simple rank of correlations. For starters, replacing the LASSO component of our model with a linear model leads to very poor performance, even when this step is preceded by a PCA (10-fold CV $R^2 = 0.02$; compared to 0.72 for original model). Second, if we use the 100 "top-correlated" genes as a feature selection for our LASSO based model or for a normal linear regression model, the cross-validation accuracy is substantially lower than using the top 100 features from our original LASSO based model. We also ran this test with the top 100 genes from the PLS model, and the results have been reproduced below, and added to Supplementary Figure 3.

Together, this indicates that non-linear expression patterns along the longitudinal axis make a significant contribution to the prediction of location position. Additional information on the different types of expression patterns contributing to the prediction model can also be found in the Responses to Reviewer 2 Comment 2 and Reviewer 3 Comment 4.

Reviewer #2 (Remarks to the Author):

Vogel and colleagues present an interesting set of findings pertaining to the molecular organisation of the human hippocampal long-axis. An excellent, concise introduction is followed by several methodological approaches to characterising long-axis organisation based on data from six deceased human donors from the Allen Human Brain Atlas. The location of hippocampal tissue samples can be correctly ascribed to their position of origin along the long-axis with accuracy. Notably, the expression pattern of only a fraction of the genes studied (100 genes of 58,692) was required to reach high accuracy. Of these 100, Random Forest Regression picked out just a handful that was important for prediction, with some genes showing differential predictive value for different subfields.

The results of the first analysis, a regularized PCA approach, are shown to converge with a second approach (another regression method based on dimension reduction; partial least squares regression). The authors then looked further afield from the hippocampus and derive an index of similarity between the genomic signature of the hippocampal longitudinal axis and that of samples from the rest of the brain (called HAGGIS). Sampling is obviously more sparse, but there is some overall trend for differences between anterior vs posterior hippocampus-associated gene expression. The midline effects are most striking, but it is perhaps difficult to understand the significance of the similarity between anterior hippocampal genetic signature and that of the extent of the dorsal aspect of the brainstem. HAGGIS from each sample was then regressed against other measures – resting-state functional connectivity, structural covariance, and glucose PET comparing AD vs frontotemporal dementia derived from open access databases – each showing a positive effect. The limitations section covers the obvious short-comings of the work, such as sample size.

These results are an important addition to a growing literature on the genetic architecture potentially underlying hippocampal long-axis organization that has until now been studied in the rodent. I have some comments for the authors to address, in order of importance.

1) - With respect to interpretation of results in terms of the AT-PM dissociation, I am not sure how well the data fits this theory. AT-PM was proposed to account primarily for perirhinal/parahippocampal dissociations. The AT system comprises lateral orbitofrontal and ventral temporopolar cortex and amygdala; while the PM system includes posterior cingulate, precuneus, angular gyrus, ventromedial prefrontal cortex. Thus, there are key anatomical differences that should not be overlooked, both in terms of inconsistencies and omissions. For the former, the vmPFC and angular gyrus are linked via HAGGIS to anterior hippocampus (predicted to be posterior and part of PM system). For the latter, how can the prominent brainstem and cerebellar relationships with anterior and posterior hippocampus, respectively, be reconciled with this account?

A linear division between brain areas – following the axis of the brain stem – seems like the most parsimonious segregation.

Response: We thank the Reviewer for these thoughtful comments. Our objective with these analyses was not to “test” the AT-PM system, but rather to point at similarities between the AT-PM system and this hippocampal genomic signature, which has been derived completely independently. The similarities are alluring not only with regard to spatial overlap, but also with regard to the behavioral contributions highlighted in Figure 6. However, we agree with the Reviewer that there are important differences as well. The reviewer for example mentions the

vmPFC associated with the PM system, but associated with the anterior hippocampus via HAGGIS. This is also reflected in some PM system behaviors (e.g. memory) being more associated with the anterior hippocampus. However, we also find the vmPFC more connected with the anterior hippocampus and more associated memory, which reproduces findings from other recent large studies examining functional connectivity along the longitudinal axis of the hippocampus (Vos de Wael et al., 2018 PNAS; Chase et al., 2015 Neuroimage). These results actually do not agree with the AT-PM arrangement, but they are concordant with HAGGIS (at least in the anterior DMN). We have now dedicated two paragraphs of the Discussion (In 481-510, reproduced below) and two supplementary figures (S6 and S11) to these and other comparisons between HAGGIS and the AT-PM. See also our response to Reviewer 3 Comment #5, which also touches on discrepancies between HAGGIS and other cortico-hippocampal systems.

The reviewer has also suggested that a linear division following the posterior axis of the brainstem (such as the one visualized below) might better explain the data than an existing biological concept, such as the AT/PM system.

By drawing such a line, we can divide the whole brain into “Anterior” (red) or “Posterior” (blue). We can then see how well a positive or negative HAGGIS value discriminates these two classes, and in doing so, find about a 72% classification rate (see below).

Most of the disagreement stems from the cortex, but some brainstem structures (mostly in the pons) have a negative value for HAGGIS, and some cerebellar structures (namely the deep cerebellar nuclei) have positive values for HAGGIS. For a comparison, we used information from Ranganath & Ritchie 2012 to classify samples into AT or PM system regions using the gold-standard

anatomical labels that come with the Allen Brain Atlas dataset. Specifically, perirhinal, amygdalar, temporal polar and orbitofrontal samples were classified as AT, whereas parahippocampal, mammillary body, anterior thalamic, precuneus, angular gyrus, medial prefrontal and posterior cingulate samples were classified as PM. Again looking at how HAGGIS discriminates these classes, we actually find a nearly identical discrimination to the simple linear division above – about 72%:

Based on this information, a linear divide along the dorsal axis of the brainstem is a less attractive summary of the HAGGIS pattern, as it does not discriminate samples any better than a biological explanation with more concrete theoretical underpinnings.

However, the Reviewer does raise the important point that the brainstem and cerebellum seem to be highly differentiated by HAGGIS, and these structures are not covered by the AT-PM hypothesis (nor are they addressed by the dual origin hypothesis). On this point, one can only speculate, but one explanation might be related to the organization of neural development. Differentiation of brain structures begins early in the developmental process of the neural tube, where early versions of major structures are arranged along either the ventral or dorsal plate (le Dréau & Martí, 2012, Developmental Neurobiology). It is possible that the gene expression signatures differentiating these anterior and posterior cortical brain regions may be partially a consequence of differential position along the neural tube during development, specifically relating to the ventral and dorsal plate. This is hard to test in the cortex because differentiation of cortical regions may occur later in development and as a consequence of a more complex diversity of gradients. However, position of subcortical structures along the ventral and dorsal plate is fairly well understood (Gilbert, 2000). To test this theory, we looked at whether HAGGIS can differentiate structures developing on the dorsal (Alar Medulla, Cerebellum, Tectum, Thalamus) or ventral (Basal Medulla, Pons, Tegmentum, Hypothalamus) plate. We find that, using a value of 0 as a reference like we did above, HAGGIS discriminates whether subcortical structures develop along the dorsal or ventral plate with 73% accuracy (see below). This is similar accuracy to AT-PM classification above, but using a completely non-overlapping set of subcortical tissue samples.

HAGGIS was not specifically designed to be associated with the AT-PM system, nor development along the neural tube, it is simply exploring variation in gene expression along the longitudinal axis of the hippocampus. That this variation also relates to completely separate systems and theories perhaps suggests a relationship between these systems and variation along the longitudinal axis of the hippocampus, as well as with one another. We do not mean to suggest HAGGIS explains these other systems, we are simply drawing empirical comparisons between them, and pointing out that genomic patterns are upstream of behavioral systems. We have tried to make these points more clear in a new paragraph in the Discussion (reproduced here):

The genomic gradient we have identified may be pertinent to the origins of some of the cortical specificity described above, however there are also a number of discrepancies worth note. For example, functional connectivity and structural covariance patterns between the hippocampus and salience and posterior default mode network regions were inconsistent with what would be expected given genomic similarity to the hippocampus (Supplementary Fig S6). Related, while the posterior DMN showed greater genomic similarity to the posterior hippocampus as would be predicted by the AT/PM system the medial prefrontal cortex was much more associated with the anterior hippocampus. In addition, regions meta-analytically active during episodic and autobiographical memory and encoding/retrieval tasks showed a genomic profile more similar to the anterior hippocampus. However, in our own resting-state fMRI analysis utilizing information from 1000 individuals, DMN structures were more connected to the anterior than the posterior hippocampus. These findings coincide with results from a previous large-scale meta-analytic coactivation study of differentiated hippocampal function along its longitudinal axis [15], and are consistent with a recent study observing changes in functional connectivity across the longitudinal axis of the human hippocampus [57].

Regarding other discrepancies, we should note that none of the previously described cortical models (Dual Origin, AT/PM, etc) include the cerebellum or brainstem, structures showing prominent divergence along the hippocampal gradient in question. This divergence may suggest that the molecular gradients defining anterior-posterior divergence in the cortex define similar divergence in subcortical structures, and point

perhaps to dorsal and ventral plate patterning during neural development [30]. While we have observed much agreement between HAGGIS and a number of other gradient-like neural organization patterns, the disagreements (summarized in Supplemental Figures S6,S11) leave much to be elucidated about how developmental and environmental molecular signals

2) - A conceptual point to make is that there appear to be a combination of effects here, with some genes expressed almost selectively in posterior vs anterior regions, while others show more of a graded pattern. This is clear from Fig 2B (note that the x-axis should be labelled in this figure). The authors are aware of this, and dedicate Supp Fig 4 to show graded organisation. Given that gradients are a core feature of the manuscript, I would consider including this figure in the main text. The co-existing patterns of segregated vs graded gene expression should be stressed, as it concurs with rodent findings.

Response: The Reviewer makes an excellent point about emphasizing the diversity of expression patterns of genes across the longitudinal axis. While Supplementary Figure S4 (now S7) is dedicated to emergent gradients in functional connectivity and structural covariance, we agree that an explicit analysis addressing genomic gradients should be added to the Main Text.

We performed Spectral Clustering across genes from Sets 1-4 (all genes found to be relevant to axis position), finding 14 different patterns of anterior-posterior expression across the longitudinal axis. As the Reviewer anticipated, linear, non-linear, stepwise and mixed patterns appeared, likely as a result of interplay between various graded and segregated patterns (visualized below in what is now Supplementary Figure 3):

In response to Point #4 by Reviewer 3, we also show that Set 1 was composed of a higher proportion of strictly linear gradients, and that even Set 1 probes within non-linear expression clusters were more linear than average. This information is visualized in subpanels C and D added to Main Text Figure 2, (visualized in the excerpt below). See response to Reviewer 3 Comment 4 for additional information.

We have integrated the above analysis to the Methods (Section 8.6) and Results (Section 2.2, In 155-168). And we have added an X-axis to Figure 2B, with apologies for the oversight.

3) - In the abstract the authors state that “posterior parieto-occipital and cerebellar regions involved in spatial cognition, selectively vulnerable to Alzheimers disease,..” I would not say that these regions are selectively vulnerable to AD. Taking the cerebellum as an example. Regardless of whether one thinks that the cerebellum is an “innocent bystander” in AD or not, or whether it is less affected by AD pathology than other regions, the cerebellum is also affected in FTD. Indeed, the paragraph in the discussion (from line 467) dedicated to AD and FTD is, in my opinion, too speculative. After all, if AD pathology is considered to begin within locus coeruleus and transentorhinal cortex, these are areas showing genetic similarity with anterior hippocampus. I appreciate that this comes down to a question of the primary pathogenic process in question, but tau pathology in AD has not been considered in the authors' discussion.

Response: The Reviewer makes several important points, many of which we agree with. Our purpose for highlighting the contrast between AD and FTD was to offer another (perhaps more downstream) example of variation along the hippocampal axis, in order to pair it against HAGGIS. We agree it is perhaps misleading to suggest the cerebellum or occipital lobe is “vulnerable” to AD. Our data show that AD patients show greater neurodegeneration in certain regions than the age-matched FTD patients, but this is different from saying these regions are not vulnerable to FTD. However, just as with the ATPM system, we are not suggesting a complete overlap between comparisons to HAGGIS. We are rather suggesting that HAGGIS represents yet another phenomenon varying

along the A-P axis of the medial temporal lobe that appears to be related to a pair of distinct cortical systems, which are themselves largely overlapping across phenomena. However, we do concede that the cerebellum and brainstem have largely not been described in previous theories of hippocampal A-P variation, and the cerebellum certainly is not thought of as a region vulnerable to AD. We have reworded the abstract and rewrote part of the discussion to avoid misleading language in relation to this topic (see below).

The Reviewer also points out that tau pathology is not mentioned in our Discussion, and that this pathology originates in brainstem and entorhinal regions that appear to be more genomically associated with the anterior hippocampus. An interesting paper was recently published suggesting tau pathology may be more related to AT system regions, while amyloid (the other hallmark pathology of AD) is associated with PM regions (Maass et al., 2019 Brain). These are interesting points, but we are not measuring molecular pathology with our analyses. Instead, we have highlighted regions that show empirically greater neurodegeneration in FTD vs AD (and vice versa), irrespective of the underlying pathology. Tau is not specific to AD and is the primary pathology in some variants of FTD, and yet the (relative) regional macroscopic degeneration (atrophy/hypometabolism) patterns of these diseases differ. Neurodegenerative disease is not the main topic of the paper, and a more in depth analysis is probably out of scope. To address the Reviewer's points, we have rewritten the discussion of this topic entirely, and we have been more careful with our wording throughout the manuscript. Here is the new paragraph in the Discussion, In 591-614:

Data from multiple studies support a specific role for the longitudinal axis of the hippocampus in AD and FTD [28, 35]. Our data support this notion, suggesting that regions more vulnerable to FTD than AD share a more similar molecular profile to the anterior than posterior hippocampus, and that the opposite pattern was observed for regions more vulnerable to AD than FTD. This relationship is far from perfect { for example, the cerebellum is not thought of as a vulnerable region in AD, nor is the medial occipital cortex (though our results showed this latter regions is empirically more impaired in AD than in FTD). Similarly, while neurodegeneration in anterior and middle temporal lobes are more severe in FTD, these regions are also vulnerable in AD. However, our results contrasting these two neurodegenerative diseases directly highlight a general divergence in vulnerability across the anterior-posterior axis of the brain, mirroring the extreme of the HAGGIS gradient. While it is tempting to wonder whether the same genes that coordinate the development of different systems also incidentally contribute to the degeneration of these systems over time, post-hoc analyses suggest that factors associating with disease vulnerability may be downstream from those factors associated with anatomical brain development (Fig S10). Although little can be extrapolated from our data about the potentially dissociated role of specific proteins in AD and FTD, we provide evidence for distinct molecular properties that characterize the dissociated hippocampo-cortical systems vulnerable to each of these two diseases. The implicated genes and proteins may provide promising candidates for more targeted

studies of their role in disease-specific neurodegeneration.

4) - I may have missed this somewhere, but demographics for the post-mortem individuals (e.g. age, gender, cause of death, post-mortem interval) should be provided.

Response: We had not included this information as it is available on the Allen Brain website and has been reported in previous publications. However, for the sake of convenience, we have aggregated all available information about the individual donors and included it as Supplementary Table S7. This table now includes information relating to age, sex, race/ethnicity, hemisphere extracted, handedness, postmortem interval and medical information for all six donors. Donor cause of death is not available, though exclusion criteria are available on the Allen Brain website, and include brain injury or disease, epilepsy, drug/alcohol dependency, <1 hour ventilator, positive for infectious diseases, prion disease, chronic renal failure, cancer death, brain cancer, and post-mortem interval > 30 hours.

Minor points

5) - In the results section, when referring to genes previously described in studies exploring the rodent longitudinal axis, please reference the relevant studies.

Response: We agree with the Reviewer that this is an important addition to the manuscript. We have moved the original Table 1 to supplementary, and have instead included a more readable list of genes associated with our model as Table 1. This table now includes information regarding which studies mentioned which genes. See also Reviewer 3's Comment 1. We have also now included similar information for the other two gene sets, in Supplementary Tables 2 and 3, respectively.

6) Also, please report first the 6 subfields used, before describing results associated with them.

Response: We thank the Reviewer for this suggestion. We have added the following sentence preceding the results on the hippocampal subfields:

“Hippocampus samples were extracted from six different subfields as labeled by the anatomist: cornu ammonis 1-4, the dentate gyrus, and the subiculum.”

7) - P. 28 methods: “all samples across the six donors were aggregated, the effect of donor was removed from each probe using linear models (i.e. with dummy coded donor ID variables)”. Does this really “remove” the effect of

age/gender etc., or just model the linear interactions between the tested variable and these parameters?

Response: We agree that this wording could be arranged to be more specific. We have updated the text in the methods, which we have reproduced here for convenience:

“ As such, all samples across the six donors were aggregated and we regressed donor-specific effects from each probe using linear models. Specifically, we used dummy coded donor ID variables to model donor-specific patterns for each probe, and by taking the standardized residuals of this model, removed variance specifically associated to each donor from each probe. Therefore, probe values represent gene expression normalized across all samples, adjusted for inter-individual statistical differences. “

8) - Legend for Figure 1 “G) The first 50 rounds of 100-probe removal from Panel A. Inflection points were identified after removing 100, 600, and 2700 genes.” Should read Panel F

We thank the Reviewer for identifying this oversight. We have corrected the Figure legend.

9) - Fig 2, Supplementary Fig. S2 “Contribution” and “Importance” are used interchangeably and might confuse some readers

Response: We appreciated the Reviewer pointing this out. We have clarified the text in the captions of Figure 2 and S2.

Reviewer #3 (Remarks to the Author):

In their manuscript titled “A molecular gradient along the longitudinal axis of the human hippocampus informs large-scale behavioral systems,” the authors find that the differential connectivity and behavioral roles of anterior and posterior hippocampus in human is reflected in a transcriptomic signature along the longitudinal axis. This work builds on similar findings in rodent by leveraging existing human genomic and imaging data sets and shows that this hippocampal axis is a conserved feature of mammalian brain across >65 million years of evolution. This work provides a useful statistical framework (and code) for identifying specific genes contributing to expression gradients in general and will be useful to the community. A major limitation of the work is that it relies on microarray data from bulk samples that include a diversity of neuronal and non-neuronal cell types. However, the authors help mitigate this by considering dissections from different hippocampal

sub-fields separately, and the results will provide a useful comparison for future single cell RNA-seq analysis from this brain region.

Specific comments:

1) • Please report which genes that have been reported to have a dorsal/ventral gradient in mouse appear (or not) in Table 1. Are there any genes that change polarity, e.g. are dorsally and anteriorly enriched?

We thank the Reviewer for this excellent suggestion. We have revised Table 1, which now includes information on which genes have been previously reported, and by which papers. None of these genes are reported to have different polarity in the other studies – anterior genes in this study were reported as ventral in the mouse studies and posterior genes reported as dorsal, in all cases. We have also added similar information into Supplementary Tables 3 and 4, which refer to Gene Sets 2 and 3.

2) • Can you comment on whether you think these gradients are driven by changes in cell type proportions or expression within cell types? Which cell classes might be involved? You can compare to single nucleus RNA-seq from mouse hippocampus or human cortex to guide this analysis.

Response: We appreciate this fascinating suggestion, which was similarly proposed in Reviewer 1 Comment #1. For convenience to the Reviewer, we have pasted that response below.

The Reviewer's point about some of the top genes from our model marking specific cell types lends to the compelling question of whether our gene expression signature is actually pointing to variation in cell type along the longitudinal axis of the hippocampus. Single cell identification and characterization is a subject of ongoing research, and identifying cell fractions based on bulk microarray data represents a significant challenge for which no standardized and multiply validated protocol has emerged. Despite these challenges, we attempt to characterize cell-type variation along the axis of the hippocampus by combining results across two different pipelines and datasets.

We first utilize CiberSortX, an online tool developed to create signature matrices from RNAseq data, and use this information to determine cell fractions from a bulk sample of RNA or microarray data (Newman et al. 2019, Nature Biotechnology). For a reference matrix, we use single cell RNAseq data extracted from the human middle temporal gyrus (MTG) – data downloaded from the Allen Brain Atlas. 15,928 MTG cells were available, and these cells have been catalogued into 169 distinct cell types using methods that have been previously described (Hodge et al., 2019 BioRxiv). In order to increase the reliability of our analyses, we split the data into four separate sets of ~4000 cells each. For each set, we used CiberSortX to create a signature matrix from the

RNAseq data and subsequently establish an estimation of the fraction of each cell type composing each of our hippocampus samples. We then removed cell types that were inconsistent (mean $r < 0.7$) between the four CiberSortX four, since these should theoretically be similar across subsamples. Only 15 cells fit this criterion, and their average fractionation is visualized below.

After averaging the cell fractions across the four runs, we found three cell types to vary significantly along the longitudinal axis of the hippocampus after a liberal correction for multiple comparisons ($FDR < 0.1$): One astrocyte, one microglia and one excitatory neuronal cell type (see Figure below). All types became more prevalent in more anterior parts of the hippocampus. In contrast, cell type varied highly across hippocampal subfield (Supplementary Table S5).

To try to validate these findings, we used a different approach to establish cell types, inspired by Seidlitz et al., 2019 *BioRxiv*, using a different reference dataset of 35 cell types taken from Lake et al., 2018 *Nature Biotechnology*. Specifically, for each sample, we calculated a weighted mean of the expression of top genes associated with each cell type, where the weights were established through cell-type specific PCA across all samples. Using this more coarse approach, six cell types were significantly associated with hippocampal axis position after liberal corrections ($FDR < 0.1$): five excitatory neuronal types and one astrocyte subtype. All cell types but one (an excitatory neuronal type) were more prevalent toward the anterior hippocampus. However, based on comparisons of Lake et al. and Hodge et al. (Fig S4 in Hodge et al.), the excitatory cell from the CiberSortX analysis was not one of those found in the present analysis. Therefore, the only cell type to replicate across the two analyses was the astrocytic cell, and these relationships are visualized in panel B of what is now supplementary Figure S5, shown below:

Based on these analyses, we have found evidence for a weak posterior-to-anterior increase of astrocyte prevalence along the longitudinal axis of the hippocampus, replicated across two different methods with different reference samples. This finding does corroborate previous histological studies of the murine dentate gyrus (Ogata & Kosaka, 2002, Neuroscience). Despite the presence of several genes marking distinct cell types and cell classes among the top features of our axis position model, our analyses suggest that the gene expression differences along the longitudinal axis are not primarily driven by distinct cell types, and that gene expression within cell-type may be varying instead. However, due to the preliminary nature of these analyses and the limitations presented by our data and methods, our conclusions remain tentative and would require more thorough and dedicated study to confirm.

We have summarized our methods, findings and interpretations in new sections in the Methods (Section 8.6), Results (Section 2.3) and Discussion (In 535-545) sections, respectively. Additional information can be found in Supplementary Table 5.

3) • You identified thousands of genes that appear to contribute to longitudinal position but only highlighted a small number of genes from the first set and that emerged from the Random Forest analysis. Do you think that only a small number of genes are relevant for setting up the appropriate connectivity and lead to cell type specific susceptibility to neurodegenerative disease? Please add more on this topic to the discussion.

Response: We agree with the Reviewer that we need to make this point more explicitly. We do not feel that the small handful of genes we have highlighted are sufficient for the processes associated with longitudinal axis development in the hippocampus, nor would we have the data to support such a claim. Rather, we have shown that a small handful of genes are sufficient for statistical purposes to predict the location of samples extracted from the hippocampus, due in large part to the linear expression gradient of these particular features across the axis. We can use this information to highlight candidate genes that could be relevant to axis development. The fact that our data replicates in a separate set of samples extracted from the prenatal human hippocampus helps to support the link between our findings and the natural phenomenon in question, as does the role some of our top genes play in A-P pattern expression in the brain, such as NR2F2, SST and PVALB. However, we have added some information into the Limitations to make this point absolutely clear:

With regard to interpretations, an important point to note is that, while only 100 genes were sufficient for statistical characterization of the hippocampal long axis, nature likely requires coordination among many more genes.

4) • Fig. 2B – Most of these genes appear to have relatively smoothly increasing anterior/posterior expression. If you correlate gene expression with anterior/posterior position, do genes with strong correlations all appear in your LASSO model? Are there genes in sets 2-4 that show low correlations with A/P position? For example, are there genes with high expression in the body and low expression in head and tail?

Response: We thank the Reviewer for posing these interesting questions, which converge in some part with Reviewer 1 Comment #6, and in another way to Reviewer 2 Comment #2. To summarize, 26 of the top 100 probes ranked by correlations with anterior/posterior position are also present in Gene Set 1 (i.e. the top 100 features from our model). These top 100 correlation-ranked features are listed in Supplementary Table S1, and the overlap with the top features of our model is visualized in Supplementary Figure 4D. However, these features lead to worse prediction models compared to our original model. More information can be found in the text, and in response to Reviewer 1 comment #6.

Regarding whether sets 2-4 showed lower correlations with axis position, and whether features were included that were hyperexpressed specifically in the body, we partitioned patterns of expression using a Spectral Clustering algorithm (please see also our response to Reviewer 2 Comment 2). This process identified fourteen distinct patterns of anterior-posterior expression, visualized below in what is now Supplementary Figure 3:

We would first like to point out Clusters 9 and 7 (2nd column). Probes in Cluster 9 (78/5000) tend to have lower expression in the body compared to the head or tail. Probes in Cluster 7 (153/5000) appear to have peaks between the extremes and midpoint of the axis. In addition, almost every expression pattern features an inflection point somewhere along the body of the hippocampus. It is possible that the model attunes to the specific locations of these inflection points to increase the precision of its predictions.

It is also worth noting that Set 1 has a higher proportion of highly linear expression gradients (clusters 3 and 6; first row) compared to the other gene sets (see below, subpanel C of the new Main Text Figure 2). Meanwhile, as the Reviewer anticipated, Sets 2-4 have a higher proportion of non-linear expression patterns.

Finally, it can be appreciated in the Figure above, as well as the Figure below (subpanel D), that Set 1 probes belonging to less linear clusters tend to have more linear expression patterns, compared to probes from other gene sets.

In all, we can conclude that a number of expression patterns are present, perhaps influenced by a combination of mixing graded and segregated molecular gradients. Many but not all of the probes in Set 1 are expressed in a pattern highly correlated with longitudinal axis position. Many probes with highly linear expression gradients are present in Sets 2-4, though the proportion of these probes are lower, and the models must rely on more probes with more diverse expression patterns to achieve comparable, or in the case of Set 4, inferior results.

5) • Fig. 4A – Sensorimotor cortex appears to have different functional connectivity and disease involvement than predicted by HAGGIS. Can you comment on some of these regional discrepancies and how one can interpret them?

Response: The Reviewer brings up an interesting point. For each modality, we have created “residual maps”, that capture the regional discrepancies between HAGGIS and expression of each modality. We have visualized these residual maps below, thresholded at 1 standard deviation to highlight the regions of greatest discrepancy. We also visualize regions that are divergent across all modalities. We have adjusted the sign of the residuals such that negative residuals showcase disagreement in directionality, whereas orange regions show disagreement in magnitude but not directionality.

The greatest and most frequent discrepancies were actually mostly in areas of apparent directional convergence between the maps, for example the medial orbito-frontal cortex. While this region is both more genomically similar to the anterior than posterior hippocampus, and is more functionally connected to the anterior than posterior hippocampus, the magnitude of these two comparisons differs substantially. The same can be said for the temporoparietal regions relating to Alzheimer's disease. The directionality is consistent but the magnitudes differ greatly.

There are of course some exceptions. For functional connectivity, default mode network (posterior cingulate and angular gyrus) and salience network (anterior insula, anterior cingulate) regions diverge from HAGGIS. For structural covariance, the posterior cingulate, anterior insula, and occipital lobe are divergent. And for disease, the sensorimotor cortex is divergent. Other regions that appear divergent may actually be an illusion of the color scale, which switches colors at 0. This may cause regions hovering around 0 to appear more different than they are. But we prefer to keep the figures this way, to clearly distinguish those regions more posterior-associated from those more anterior-associated.

We have added the above figure to the manuscript as Supplementary Figure 6, and at the Reviewer's request, discuss some of the observed divergences in more detail in the Discussion (Ins 481-511), which has also been reproduced above as part of the response to Reviewer 2 Comment #1.

6) • Fig. 4B – It is interesting that Set3 is more predictive than Set1 for disease. Is there any GO enrichment for these genes that help explain this result?

Response: This is a very interesting question that would probably benefit from a more focused analysis, which may be outside of the scope of this paper. However, based on our gene ontology analysis, many of the genes in Set 1 are associated specifically with anatomical developmental processes and axon guidance. In contrast, genes in Sets 2 and 3 are associated with a diversity of processes and functions. Vulnerability to the neurodegenerative diseases we study here may therefore be associated with factors downstream of the developmental processes that bring about the hippocampal longitudinal axis.

*In order to address this hypothesis, we returned to the clusters developed from our gene ontology analyses, where genes were clustered based on shared functions (Section 8.5, 2.2). For each cluster, we found the weighted mean gene expression separately for posterior-associated and anterior-associated genes, where the weights were defined by cluster centrality, creating “cluster scores”. We then performed correlations between mean cluster score and relative vulnerability to FTD or AD, across all brain samples. For each weighted mean, we scrambled the weights 100 times and performed correlations once again to find a null distribution. We plotted the upper 95% confidence interval of this distribution to determine if the associations we found were greater than chance. Each of these associations is plotted below, where gray bars represent the 95% confidence interval of the null distribution for that cluster, and an * indicates the association is greater than the null distribution. Set2 is on the left, with Set3 on the right.*

Clusters associated with processes similar to those enriched in Set1 (i.e. Set2 C1; Set3 C4) were not related to disease vulnerability, lending some support to our hypothesis. Interestingly, certain clusters seem to be driving the association with disease vulnerability, particularly amine activity and transport and phosphorylation, as well as some contribution from genes associated with hormonal signaling, luteinizing hormone activity, neuropeptide activity, serotonin binding and vascular growth factor activity.

This analysis has been added to the manuscript as Supplementary Figure 10, and along with comments from Reviewer 2 Comment #3, the findings are discussed in a newly written paragraph related to the association with HAGGIS and disease vulnerability (In 591-614, reproduced above in response to Reviewer 2 Comment 3).

REVIEWERS' COMMENTS:

Reviewer #1 (Remarks to the Author):

I would like to thank the authors for thoroughly responding to the questions and comments from my initial review. I have no further questions and recommend publication.

Reviewer #2 (Remarks to the Author):

The authors have provided a thorough and well thought out response to my previous comments.
Bryan Strange

Reviewer #3 (Remarks to the Author):

All of my comments have been addressed, and I support publication.